# Photoelectrochemical water splitting cells at elevated pressure using BiVO$_4$ and platinized III-V semiconductor photoelectrodes

Feng Liang [1,2] ✉, Heejung Kong [1,3], Diwakar Suresh Babu [1,3], Roel van de Krol [1,3] ✉ & Fatwa F. Abdi [1,4] ✉

Direct production of pressurized green hydrogen via photoelectrochemical water splitting reduces the need for mechanical compression and mitigates bubble-related losses. However, existing demonstrations have been limited to atmospheric pressure. Here, we bridge this gap by designing, constructing, and testing a high-pressure flow cell for photoelectrochemical water splitting using two configurations. In a back-illuminated BiVO$_4$-based photoelectrochemical cell, increased pressure suppresses bubble evolution and alleviates photocurrent saturation under concentrated sunlight: at 10 suns, the photocurrent rises from 3× at 1 bar to ~7× at 5 bar. Direct operando imaging of the electrode surfaces confirms that this improvement comes primarily from suppressed bubble evolution. Conversely, a front-illuminated platinized triple-junction III-V-based photoelectrochemical cell shows limited pressure dependence up to 8 bar due to its dispersed catalyst and long carrier diffusion length. These findings highlight the differing response of photoelectrochemical devices to elevated pressure and demonstrate a viable pathway toward scalable, high-pressure solar-driven hydrogen production.

As the transition from fossil fuels to sustainable energy sources accelerates, solar water splitting emerges as a promising strategy to store intermittent solar energy in the form of hydrogen and other energy-dense molecules that are compatible with existing infrastructures[1]. Three main types of solar water splitting systems have been demonstrated: (1) photovoltaic-electrolyzer (PV–EC) configurations, where a PV cell is coupled with an electrolyzer, (2) photoelectrochemical (PEC) cells, which integrate light absorption and electrochemical reactions in a single device[2], and (3) photocatalytic (PC) systems, where the reactions occur at the surfaces of suspended or immobilized particles. For PC water splitting systems, near-unity conversion yields have been achieved under UV irradiation[3]. Moreover, the feasibility of scale-up to larger areas using photocatalyst sheets was demonstrated with a 100 m$^2$ outdoor prototype panel reactor system[3–5], although the maximum solar-to-hydrogen (STH) efficiency was only ~0.76%[5]. A slightly higher STH

efficiency of 1.21% was recently demonstrated with a PC water splitting system that utilizes an I$_3^-$/I$^-$ redox mediator. Employing MoSe$_2$-loaded halide perovskites (CH(NH$_2$)$_2$PbBr$_{3-x}$I$_x$) for H$_2$ evolution and NiFe-layered double hydroxide-modified BiVO$_4$ for O$_2$ evolution, this 700 cm$^2$ system demonstrated stable operation over one week under natural sunlight[6]. PV-EC systems are more technologically mature and show high STH efficiencies, but suffer from thermal losses when PV panels operate at higher temperature[7,8]. These losses can be mitigated by thermally coupling the PV and electrolyzer cells, where water acts as both a coolant for the photoabsorber and as an electrolyte[9,10]. This has resulted in STH efficiencies greater than 20% at the device level when using solar concentration[11], and up to 8.5% without solar concentration[12]. However, more than 30% of the generated heat is still lost[11]. PEC systems offer an alternative by directly integrating semiconductors and electrocatalysts in one single device, allowing for synergistic effects, such as improved

[1]Institute for Solar Fuels, Helmholtz-Zentrum Berlin für Materialien und Energie GmbH, Berlin, Germany. [2]School of Mechanical Engineering, Xi'an Jiaotong University, Xi'an, Shaanxi, China. [3]Institute for Chemistry, Technische Universität Berlin, Berlin, Germany. [4]School of Energy and Environment, City University of Hong Kong, Kowloon, Hong Kong SAR, China. ✉e-mail: feng.liang@xjtu.edu.cn; roel.vandekrol@helmholtz-berlin.de; ffabdi@cityu.edu.hk

reaction kinetics and mass transport, through system heating from thermalized and sub-bandgap photons[13–17]. Recent laboratory-scale PEC devices (~1 cm$^2$) have already demonstrated STH efficiencies of ~20%[18,19], highlighting significant potential for efficient solar-to-fuel conversion.

With record-high STH efficiencies demonstrated, efforts are now focused on scaling up PEC water splitting systems. In general, PEC water splitting can be scaled by increasing the photoactive area per device[20], by deploying a greater number of devices[2,20,21], and/or by utilizing higher solar concentrations[5,9,22]. Among these strategies, the use of higher solar concentrations has been particularly promising for achieving economically viable devices with high power densities[9,11,23]. However, the photocurrent increases sub-linearly with solar concentration, causing the performance to plateau[23–25]. This sub-linear increase has also been reported in studies using monochromatic light under low irradiance ($< 0.4\,kW\,m^{-2}$) to investigate charge transfer mechanisms[26–28]. While the exact cause is not always clear, several factors may be responsible, including ohmic losses (from the substrate, the electrolyte, or from gas bubbles)[23,25], surface recombination[28], bubble-induced light scattering[29–31], and blocking of the PEC active area by bubbles[31–33]. Understanding and disentangling these effects is essential for optimizing system performance and enabling the efficient scaling of solar water splitting systems.

One approach to addressing performance saturation in PEC water splitting under concentrated solar irradiance is to operate the system at elevated pressure. This is especially relevant as gas bubbles can limit the performance when their volume fraction in the electrolyte becomes too high. The volume fraction of the gas bubbles can be effectively controlled by adjusting the operating pressure[34]. We recently reported multiphysics simulations showing that bubble-induced performance losses can be minimized by increasing the operating pressure of PEC devices to ~6–8 bar[35]. Furthermore, producing hydrogen directly at high(er) pressure aligns well with the requirements of most downstream processes (e.g., fuel cells, ammonia production, methanol synthesis), minimizing the need for energy-intensive mechanical compression[36,37]. Despite these advantages, all PEC water splitting demonstrations to date have been conducted at atmospheric pressure. As illustrated in Fig. 1a, this represents a clear research gap that needs to be addressed.

In this study, we experimentally demonstrate PEC water splitting at elevated pressure using a custom-designed high-pressure laminar flow cell. Two different PEC configurations were investigated. In the first configuration, a back-illuminated BiVO$_4$ photoanode was employed in a PEC cell operating at pressures up to 8 bar under concentrated solar irradiance (up to 10 suns). At 10 suns, a significant photocurrent increase of ~40% was achieved when operating above 5 bar instead of at atmospheric pressure. This improvement is primarily attributed to the suppression of gas bubble formation on the photoelectrode surface. In the second configuration, a platinized triple-junction (3 J) III-V photoelectrode was used in an integrated PEC cell. Interestingly, even under front illumination, we observed only a ~10% photocurrent reduction due to gas bubbles, and increasing the operating pressure had minimal impact on performance. We attribute this to the long carrier diffusion length in the III-V semiconductor and the high hydrophilicity of the surface, which facilitates bubble detachment. Furthermore, stability measurements reveal that an increase in operating pressure has no significant impact on the degradation of photoelectrode materials. Our quantitative findings reveal that the effects of increasing operating pressure on PEC cell performance are configuration-dependent, thereby offering valuable insights for scaling PEC water splitting devices.

## Results

### High-pressure flow cell for photoelectrochemical water splitting
A schematic of the custom-designed high-pressure PEC cell is shown in Fig. 1b, an exploded view of the cell design is shown in Fig. S1, and

digital photographs of the setup can be found in Figure S2. Further details on the cell design and construction are available in the Methods section. High pressure operation was accomplished by supplying compressed gas into the cell while regulating the outflow rate of gases with a back-pressure controller (Bronkhorst High-Tech, uncertainty: 0.2%). Note that this approach to regulate the pressure was chosen because it offers easy control and experimental convenience in a lab-based setting; in practical applications, PEC water splitting cells can self-pressurize through gas production during operation, requiring only a pressure relieve valve as a control mechanism. Prior to PEC experiments, the setup was tested for safe operation up to 8 bar(a); higher pressures have not been tested to comply with current safety guidelines. For illumination, AM1.5 G ($100\,mW\,cm^{-2}$) simulated sunlight was used with an optional Fresnel lens to achieve solar concentrations up to 10 suns (see Supplementary Note S1 and Fig. S3 and Fig. S4).

Our high-pressure PEC cell features two key innovations. First, it enables laminar electrolyte flow between two parallel electrodes at elevated pressure, facilitated by an optimized flow distributor (as shown in Fig. S2c). Second, the cell includes two observation windows (see Fig. S1), allowing real-time in situ visualization of the evolving gas bubbles from multiple angles. To confirm that the electrolyte flow profile is laminar, we conducted particle imaging velocimetry (PIV) measurements. A detailed description of the measurements is provided in Methods and Fig. S5. Colormaps of electrolyte velocity at different operating pressures are shown in Fig. 2a. At a constant flow rate of ~4.6 mL s$^{-1}$ (used for most of our experiments), the electrolyte flow field remains uniformly distributed between the two electrodes, with minimal variation across different operating pressures. Characteristic parabolic velocity ($\mathbf{u}_y$) profiles are observed at 1, 3 and 5 bar (see Fig. 2b), with an average velocity of ~1 cm s$^{-1}$. Beyond liquid flow visualization, our versatile cell design also enables monitoring of gas bubble dynamics from different viewpoints using multiple cameras (as shown in Fig. S6) at elevated pressure.

### Back-illuminated single photoanode cell configuration
We first utilized the high-pressure flow cell (HPFC) described above to investigate BiVO$_4$ photoanodes at elevated pressure and concentrated back-side illumination. The experimental setup consisted of a BiVO$_4$ working electrode, a platinum mesh counter electrode, and an Ag/AgCl reference electrode. Undoped BiVO$_4$ photoelectrodes (without co-catalysts) were fabricated via a previously reported electrodeposition method[38], as detailed in the Methods section. The structural and optical properties of the synthesized BiVO$_4$ are consistent with those of typical electrodeposited BiVO$_4$[38], as confirmed by scanning electron microscopy (SEM), X-ray diffraction (XRD), UV-vis spectroscopy and film profilometry (Figs. S7–S8).

Figure 3a shows the linear sweep voltammetry (LSV) curves measured at atmospheric pressure (1 bar) under various solar concentrations, with a final measurement at 1 sun (dashed curve) to confirm reproducibility. As expected, the photocurrent increases with higher solar irradiance. However, the increase is less pronounced than anticipated; for instance, under an irradiance of ~10 suns, the photocurrent is only ~3 times higher than that observed at 1 sun. To better illustrate the correlation between photocurrent and solar concentration, we extracted the photocurrent at 1.23 V versus the reversible hydrogen electrode (RHE) and normalized it to the value at 1 sun ($\mathbf{J}_{c=x}/\mathbf{J}_{c=1}$). The normalized photocurrents, plotted as red datapoints in Fig. 3b, reveal a saturation at higher solar concentrations. This behavior is consistent with previous reports in literature. For comparison, we include the data from Vilanova et al.[23], who observed a similar trend (orange datapoints in Fig. 3b) despite significant differences in experimental conditions. Their study utilized $\alpha$-Fe$_2$O$_3$ photoelectrodes (active area ~6.25 cm$^2$) in 1 M KOH (pH = 13.6) electrolyte. A comparable photocurrent saturation was also reported by Holmes-Gentle et al. for

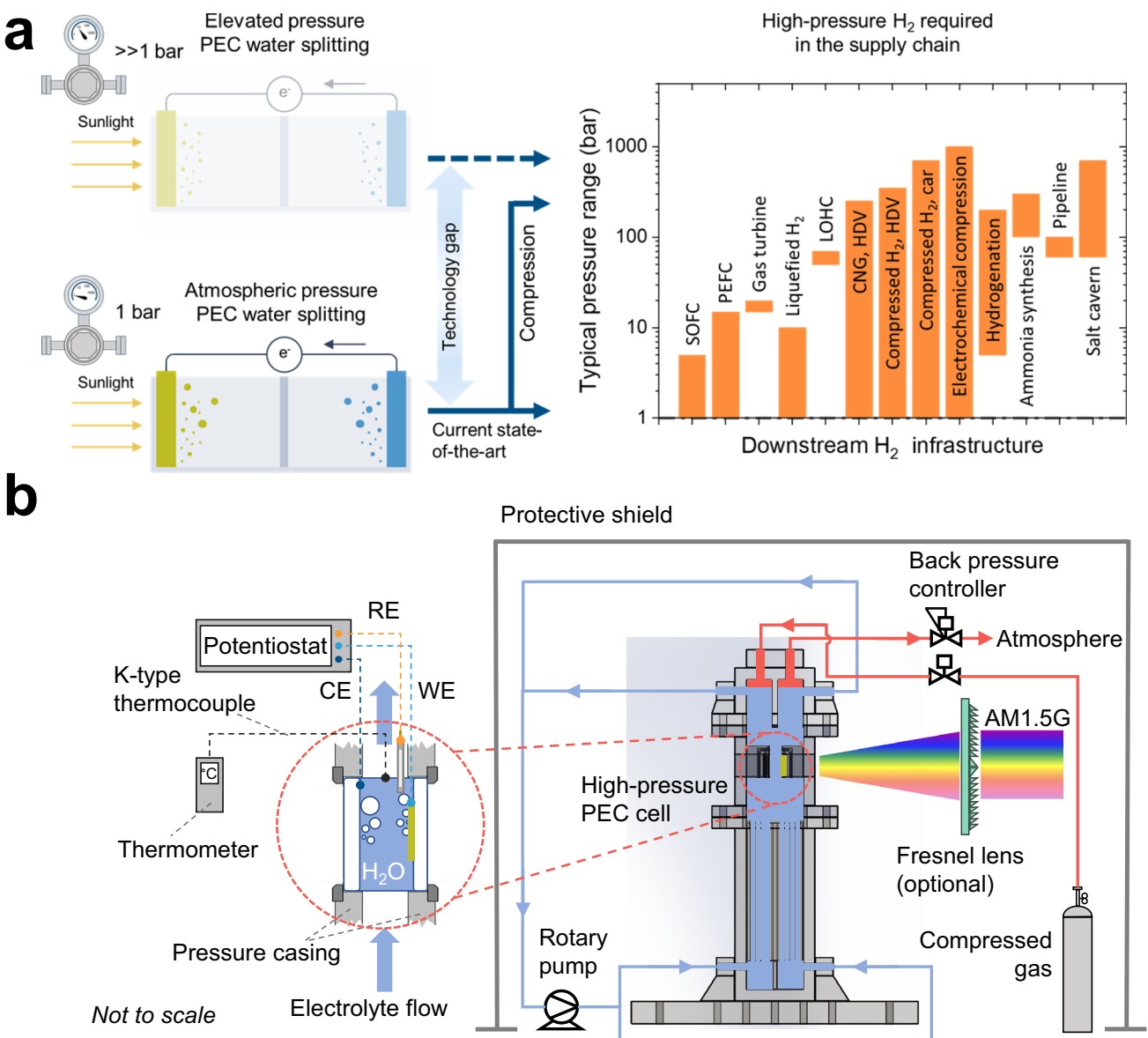

**Fig. 1 | The need to operate PEC water splitting devices at elevated pressure.** In (**a**), typical PEC water splitting setups are shown on the left. The solid schematic represents systems operating at atmospheric pressure, which have been experimentally demonstrated, while the transparent schematic signifies the concept of elevated-pressure operation, which has not yet been achieved. In contrast, most downstream hydrogen applications—such as fuel cells, hydrogenation processes, and ammonia synthesis—require high-pressure hydrogen, as indicated on the right. This reveals a critical research gap between current academic PEC studies and industrial-scale requirements. **b** presents the schematic of the HPFC designed, constructed, and tested in this study. An exploded technical view is provided in

Fig. S1, and the digital photographs of the assembled HPFC are shown in Fig. S2. The electrolyte is circulated by a rotary pump, achieving laminar flow between the electrodes via an optimized flow distributor (see Fig. S2c). The system is pressurized using external gas ($N_2$ or $O_2$, depending on the reaction), and the operating pressure is regulated by a back-pressure controller. Simulated AM1.5 G illumination —with optional Fresnel lenses—enables solar concentrations up to ~10 suns. Electrolyte temperature is monitored at the outlet via a K-type thermocouple. The downstream pressure requirements for hydrogen infrastructure included in (**a**) are adapted from ref. 37. For clarity, standard fittings (e.g., nuts, connectors, PTFE tubing) are omitted in the schematic.

$\alpha$-$Fe_2O_3$ and $BiVO_4$ photoelectrodes under much higher solar concentrations (40 to 360 suns)[25].

A possible explanation for the observed photocurrent saturation is the increased formation of gas bubbles at higher concentrations. Gas bubbles are known to block active sites on the photoelectrode[39–42], reducing its effective catalytic area, and to scatter incident light[29,33], diminishing the number of absorbed photons. While bubble-induced convection may enhance local mass transfer[33,43], its effect is negligible at the relatively low current densities in this study (e.g., ~5 mA cm$^{-2}$ at 10 suns for unmodified (bare) $BiVO_4$ photoanodes without hole scavenger). Since our experiments employed back-side illumination—where light does not pass through the gas bubbles before reaching the

$BiVO_4$ photoanode—the contribution of light scattering should also be minimal. We therefore hypothesize that photocurrent saturation is primarily due to electrode deactivation associated with the increased gas bubble coverage on the surface of $BiVO_4$ at higher solar concentrations.

To test the hypothesis above, we performed the LSV measurements using an electrolyte containing 0.5 M sodium sulfite ($Na_2SO_3$) as a hole scavenger. Due to its more favorable thermodynamics and kinetics[44], sulfite oxidation replaces water oxidation, preventing gas bubble formation (see Supplementary Video 1). The normalized photocurrents ($J_{c=x}/J_{c=1}$) in the presence of hole scavenger are plotted as blue datapoints in Fig. 3b. The photocurrent increases with higher

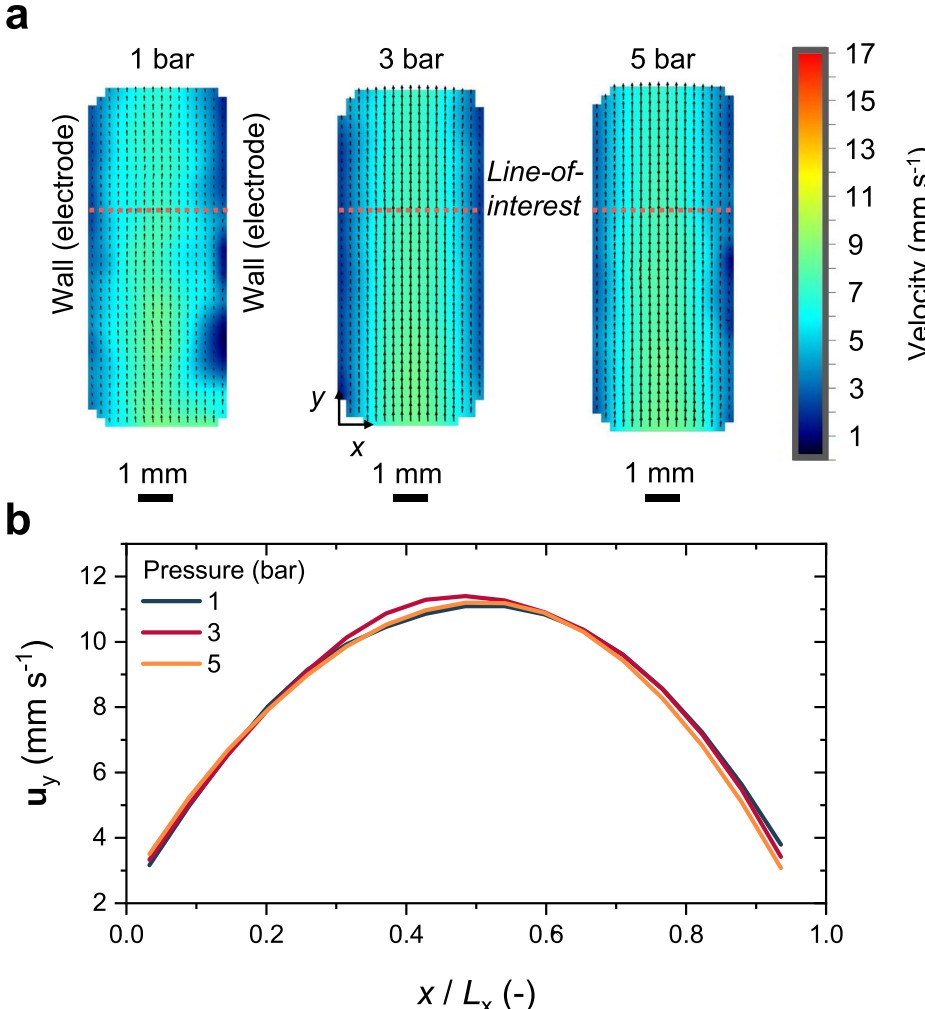

**Fig. 2 | Laminar flow profile of electrolyte. a** Representative colormaps of the liquid flow velocity in our PEC cell (Fig. 1) under different pressure. The average flow rate was fixed at ~4.6 mL s⁻¹ at all pressure. **b** Liquid flow velocity profile in $y$ direction at the line-of-interest (horizontal dashed lines in Fig. 2a). A nearly parabolic velocity profile is observed for all pressure, indicating a laminar flow is maintained between the two electrodes. The measurements were conducted at room temperature ( ~25 °C).

concentrations, and the increase is more pronounced compared to that in the absence of hole scavenger. Note that the normalization excludes the effect of reaction kinetics, as all values are expressed as multiples of their respective photocurrents at 1 sun. The absence of bubbles during Na₂SO₃ oxidation accounts for the increased normalized photocurrent compared to water oxidation. The blue-shaded area in Fig. 3b highlights the disparity, which is primarily attributed to the surface coverage of gas bubbles.

Interestingly, even in the presence of a hole scavenger, the normalized photocurrent still does not increase proportionally with solar concentration. The purple-shaded area in Fig. 3b highlights the difference between the expected one-to-one photocurrent increase with solar concentration and the observed increase in the presence of hole scavenger. This observation suggests that factors other than gas bubble coverage limit the photocurrent of BiVO₄ at higher concentrations. One possible limiting factor is mass transfer; however, this is unlikely to occur in our setup, as the electrolyte is continuously circulated at a relatively high flow rate of ~4.6 mL s⁻¹, and varying the scan rate has a negligible effect on the LSV curves (see Fig. S9). Vilanova et al. suggested that substrate sheet resistance was a primary limiting factor in their study[23], but this is unlikely to be the case for the current densities observed in our study. Using a previously established relationship[22], we estimate that the potential drop due to substrate

sheet resistance only increases from ~5 to ~20 mV as the solar concentration increases from 1 to 10 suns. In view of the relatively shallow profile of the **J**-$V$ curve (Fig. S9), this will indeed have a negligible effect on the current density. We tentatively attribute this loss (represented by the purple-shaded area in Fig. 3b) to increased bulk recombination, which has indeed been reported for BiVO₄ under higher light intensities[28,45].

We briefly note that temperature variations during the LSVs under different solar concentrations can be ignored in our study as several measures were implemented to minimize its effect. First, the electrolyte was circulated using a rotary pump with a flow rate of ~4.6 mL s⁻¹, resulting in an average linear flow velocity of ~1 cm s⁻¹ between the two electrodes (see Fig. 2 for a visualization of the electrolyte flow velocity). Second, a ~5 min resting period was included between experiments to prevent continuous heating of the electrolyte. Indeed, the electrolyte temperature variation, measured using a K-type thermocouple (see Fig. S10a), was found to be less than 2.5 °C between the highest and lowest solar concentrations (see Fig. S10b).

We then investigated how pressure elevation affects the PEC performance of the back-illuminated BiVO₄ photoanode. Higher pressures are expected to suppress gas bubble formation[35], potentially enhancing the photocurrent, especially at higher solar concentrations. Figure 4a shows the normalized photocurrent of BiVO₄ at 1.23 V vs.

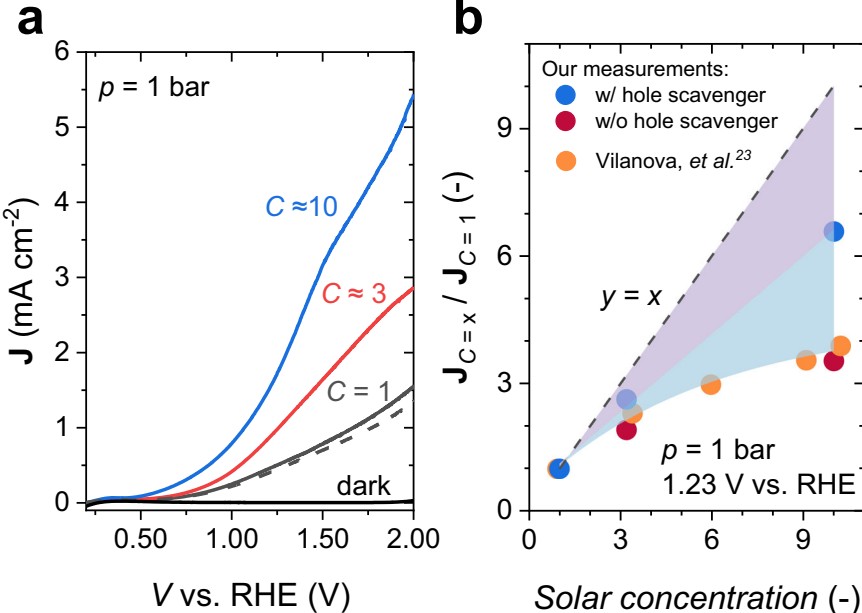

**Fig. 3 | Photoelectrochemical performance of BiVO₄ photoanodes under various solar concentrations at 1 bar. a** Linear sweep voltammetry (LSV) curves for our BiVO₄ photoelectrodes under different solar concentration at a scan rate of 20 mV s⁻¹, measured without a hole scavenger. A final measurement at 1 sun (dashed curve) was conducted to confirm reproducibility. The electrolyte used is 0.1 M KPᵢ (pH = 7 ± 0.1). No iR corrections was conducted for the reported voltammograms. **b** Normalized photocurrents ($J_{c=x}/J_{c=1}$) at 1 bar and 1.23 V vs. RHE under various solar concentrations for our BiVO₄ photoelectrodes in a 0.1 M KPᵢ solution (w/o hole scavenger, red datapoints) and in a 0.1 M KPᵢ solution containing 0.5 M Na₂SO₃ (w/ hole scavenger, blue datapoints). A reported measurement from Vilanova et al.[23] (using α-Fe₂O₃ photoelectrodes in 1 M KOH) under the similar solar concentration range is added (orange datapoints) for comparison. For our datapoints, three independent measurements were performed to ensure reproducibility, and the spread of the values is smaller than the size of the circles. The $y = x$ dashed line indicates the linear dependence of $J_{c=x}/J_{c=1}$ with the increase in solar concentration (the ideal case). The electrolyte flow rate was fixed at ~4.6 mL s⁻¹, and the electrolyte temperature was measured using a K-type thermocouple; a temperature variation of less than 2.5 °C between the highest and lowest solar concentrations was observed during the LSV measurements. The measurements were conducted at room temperature (~25 °C).

RHE at different pressures in a 0.1 M KPᵢ electrolyte (without hole scavenger). The corresponding LSVs under different concentrations are shown in Fig. S11a–c. While the photocurrent at 1 sun remains largely unaffected by pressure, the photocurrent increases significantly with pressure at higher solar concentrations. At ~3 suns, pressure elevation improves the normalized photocurrent by ~30%, and at ~10 suns, the enhancement even reaches ~50%. This pressure induced improvement in photocurrent is more clearly illustrated in Fig. 4b. At 1 sun, the photocurrent remains nearly constant at ~0.5 mA cm⁻² across the pressure range of 1 – 8 bar. At ~3 suns, however, the photocurrent increases from ~0.9 mA cm⁻² at 1 bar to ~1.2 mA cm⁻² at 5 bar. The effect is more pronounced at ~10 suns, where the photocurrent increases from ~1.7 mA cm⁻² at 1 bar to ~2.5 mA cm⁻² at 5 bar. Furthermore, as shown in Fig. S11, the onset potential of the BiVO₄ photoelectrode remains unchanged with varying operating pressure. These findings demonstrate a key advantage of operating PEC water splitting devices under elevated pressure, particularly at higher solar concentrations.

To determine if the pressure-induced enhancement in photocurrent stems from the suppression of bubble evolution (or a decrease in bubble size, see below), we performed the same measurements in the presence of a Na₂SO₃ hole scavenger (so that no bubbles are generated), and the LSVs are presented in Fig. S11d. The normalized photocurrent remained unchanged with increasing pressure for all solar concentration factors (see Fig. 4c). This indeed confirms that the increase in photocurrent with pressure, observed in Fig. 4a, b, is due to the suppression of bubble evolution. This result is consistent with direct visual observations from our PEC cell, the design of which allows in situ monitoring of gas bubble evolution on the photoelectrode from multiple angles (front- and side-view, see the schematic illustration and photographs in Fig. S6). Representative images of gas bubbles on

BiVO₄ photoelectrode at 1 bar and 5 bar are shown in Fig. 4e, f, respectively, with corresponding videos available in Supplementary Videos 2–4. At 5 bar, noticeably fewer bubbles were observed compared to 1 bar, as evidenced by the images and videos. In addition, we quantified the average bubble size ($D_{bub}$) and number of bubbles ($N_{bub}$) under different pressures and solar concentrations, as shown in Fig. 4d. Both $D_{bub}$ and $N_{bub}$ decrease with increasing pressure, especially at higher solar concentrations. These findings confirm that the suppression of gas bubble evolution is the primary mechanism behind the improved photocurrent observed at elevated pressures.

It should be noted that while an increase in operational pressure effectively suppresses gas bubble evolution on the BiVO₄ photoelectrode (see Fig. S12), the normalized photocurrent at 8 bar in the absence of a hole scavenger (~5.3) remains lower than that in the presence of scavenger (~6.8). We attribute this discrepancy to the resolution limits of our imaging setup. Although the images suggest "bubble free" conditions at 8 bar, nano- or micro-bubbles below our detection threshold may still be present and block a significant fraction of the electrode surface. To support this hypothesis, we estimated the Laplace pressure[46,47] for bubbles of varying sizes (see Supplementary Note S2 and Fig. S13). For bubbles with a radius ≤ 0.2 μm, the Laplace pressure at bubble/electrolyte interface exceeds 8 bar, meaning such small-sized bubbles are unlikely to be suppressed under our experimental conditions. The detection threshold of the bubble radius with our microscopic camera lens is about 10 μm.

Although bubble nucleation events could not be directly resolved, recent work on bubble evolution on TiO₂ micro-photoelectrodes showed that elevated pressure facilitates bubble detachment, primarily due to higher dissolved gas saturation[48]. Consistent with these findings, our measurements (Fig. 4e, f) revealed smaller and fewer bubbles under elevated pressure. This behavior can be tentatively

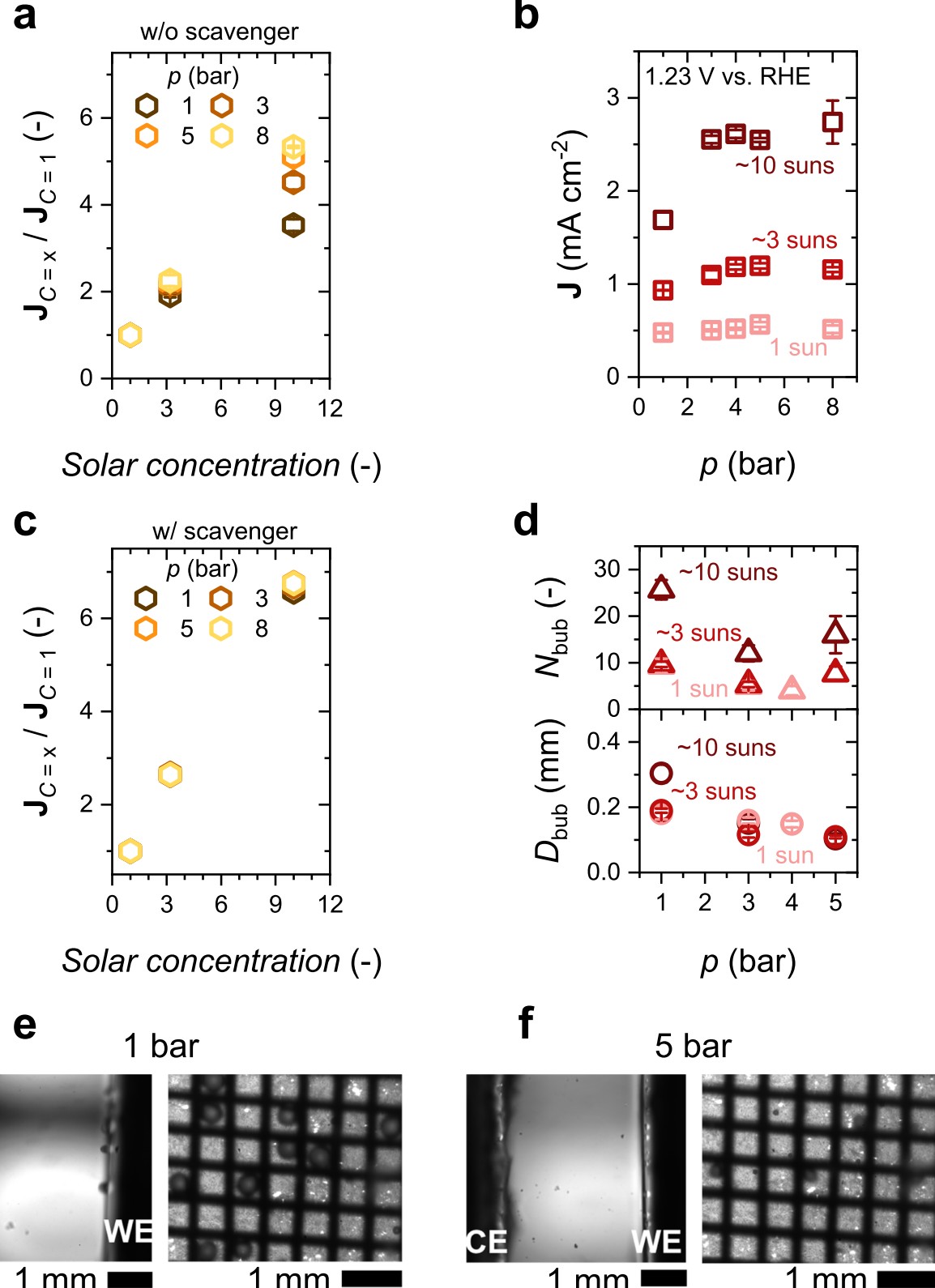

**Fig. 4 | Effect of pressure on the back-illuminated BiVO₄ PEC water splitting cell. a** Normalized photocurrent of our BiVO₄ photoelectrodes in a 0.1 M KP$_i$ solution (pH = 7 ± 0.1). **b** Photocurrent of our BiVO₄ photoelectrodes at 1.23 V vs. RHE as a function of pressure under different solar concentrations in 0.1 M KP$_i$ (pH = 7 ± 0.1). **c** Normalized photocurrent of our BiVO₄ photoelectrodes in 0.1 M KP$_i$ (pH = 7 ± 0.1) with 0.5 M Na₂SO₃ added as a hole scavenger. **d** The number of bubbles ($N_{bub}$, upper panel) in the region of interest and the average bubble diameter ($D_{bub}$, lower panel) as function of pressure for various solar concentrations.

Representative images of gas bubble evolution under (**e**) 1 bar and (**f**) 5 bar (left: side-view, right: front-view). The counter electrode (CE) is a platinum mesh, which is shown as the grid in (**e**) and (**f**), and Ag/AgCl is used as the reference electrode. The electrolyte is circulated at a flow rate of ~4.6 mL s⁻¹. The pressure in the cell is increased by introducing compressed O₂ gas. The error bars in (**a**–**d**) represent the standard deviations of at least three measurements. Scale bars in (**e**) and (**f**) represent 1 mm. The measurements were conducted at room temperature (~25 °C).

explained by the formation of a supersaturated boundary layer (SSBL) of dissolved gas near the (photo)electrode at higher pressure. Our recent simulation showed that the SSBL facilitates bubble detachment from the surface of the (photo)electrode via Marangoni convection[49]. Additional contributing factors may include an increased Laplace pressure and a larger critical radius for nucleation at higher pressures, both of which inhibit initial bubble formation. Interfacial properties, such as the gas–liquid contact angle, may also play a role in pressure-dependent detachment dynamics, but direct experimental measurement of contact angles and nano/micro bubbles on semiconductor surfaces (e.g., $BiVO_4$ photoelectrode) under elevated pressure remains technically challenging. High-resolution tools such as atomic force microscopy and spectroscopy techniques[50–60] and molecular dynamics simulation[61,62] will be essential, which are beyond the scope of this study. Furthermore, the porous morphology of our $BiVO_4$ samples (see Fig. S7) introduces additional complexity, as it may lead to random spatial distribution of nanobubbles, making their detection even more complicated.

To further elucidate the effects of pressure on gas bubble evolution throughout the whole surface of the 2 cm × 2 cm $BiVO_4$ photoelectrode, we conducted macroscopic imaging of oxygen bubbles. The experimental setup was modified accordingly (Fig. S14a) and introduced in the Methods section. At 1 bar, bubbles appeared randomly across the porous $BiVO_4$ surface (Fig. S14b), with larger bubble sizes observed near the top of the electrode; we tentatively attribute this observation to the presence of hydrostatic pressure gradients. This insight is useful for the scale-up of PEC devices, as non-uniform bubble detachment may result in spatial variations in performance. However, at pressures above 2 bar, bubble presence was already negligible (Fig. S14b). This observation is consistent with our localized observations shown in Fig. 4e, f.

To investigate whether pressure elevation introduces any effect on the stability of the back-illuminated $BiVO_4$ photoanode, potentiostatic experiments were performed with the same PEC cell at 1.23 V vs. RHE. Illumination intensities of 1 sun and 10 suns were applied. As shown in Fig. S15, the photocurrent decreases with time, indicating degradation; this is expected since our $BiVO_4$ is uncatalyzed and has no protection layer[25,63]. However, the photocurrent decrease remains similar at different pressures, suggesting that pressure elevation has minimal impact on the degradation behavior of the $BiVO_4$ photoelectrode. Moreover, no gas or liquid leakage was observed during the extended high-pressure operation, indicating reliable structural mechanical integrity and operational stability of the system.

A heat transfer model was developed to evaluate the thermal effect associated with the prolonged operation under higher solar concentration, as detailed in Supplementary Note S3 (see Fig. S16). The simulated results were validated against experimentally measured bulk electrolyte temperature, as shown in Fig. S17a. Despite its simplicity, this heat transfer model effectively captures the thermal trends observed during stability testing under ~10 suns illumination. Specifically, the measured bulk electrolyte temperature increased by approximately 5 °C at 1 bar and 3 °C at 5 bar under continuous illumination, while the model predicts a comparable rise of ~5 °C for both pressures. In addition, the simulation reveals that the maximum surface temperature rise of the photoelectrode can reach ~13 °C above ambient (Fig. S17b–d). While this value may be somewhat overestimated due to simplified boundary assumptions—namely, adiabatic treatment of the electrode sealing interfaces and the electrolyte boundary adjacent to the counter electrode—it nonetheless highlights the potential for localized heating at the photoelectrode surface. Such a temperature rise can indeed facilitate interfacial kinetics[64] and could potentially accelerate material degradation under extended operation[65]. These findings emphasize the importance of effective thermal management in high-pressure PEC systems under concentrated sunlight. Advanced cooling strategies, such as those

demonstrations performed by Haussener and co-authors[9,11,25], can be implemented to prevent photoelectrode overheating under concentrated sunlight. Importantly, the consistent trends observed across under different pressures, both experimentally and in simulation, support the validity of our comparative performance analysis under elevated pressures.

## Front-illuminated integrated PEC water splitting cell configuration

We now turn to the effect of pressure on the front-illuminated integrated PEC water splitting cell. In this configuration, front illumination is necessary due to the non-transparent contact layer on the back side of the PV cell. The photoelectrode is a platinized triple-junction (3 J) III–V (GaInP/GaAs/Ge) PV cell, prepared as described in the Methods section. The digital photographs of the photoelectrode are shown in Fig. S18. The photoelectrode functions as a photocathode, where the nanoparticulate Pt catalyst facilitates the HER ($2H^+ + 2e^- \rightarrow H_2$) and an $IrO_x/TaO_x/Ti$ mesh (mmoelectrode, China) serves as the counter electrode catalyzing the oxygen evolution reaction (OER, $2H_2O \rightarrow O_2 + 4H^+ + 4e^-$). An Ag/AgCl electrode was used as the reference. To optimize gas bubble imaging on the electrode surface, the working electrode was tilted at ~45° from the horizontal. This adjustment was necessary for the integrated PEC cell but not for the $BiVO_4$ photoelectrodes due to three reasons. First, the area of the III–V photocathode is a lot smaller (~ 1/6) than that of the $BiVO_4$ photoelectrodes. Second, the edges of the photocathode need to be protected with silicone resin to prevent direct contact with the aqueous solution (see Fig. S18). Third, keeping the photocathode tilted (e.g., at 45° from horizontal) reduces the density of bubble plume. The counter electrode (CE) was carefully placed to avoid shadowing effects from both the CE itself and from the oxygen bubbles that are generated at its surface. The 3 J III–V PV cell is made of GaInP/GaAs/Ge junctions with varying bandgaps (1.8/1.4/0.7 eV) to absorb photons from the entire spectrum, including the infrared region[66]. To ensure effective illumination of the bottom Ge junction, the experiments were conducted with a dual light source solar simulator, with a halogen lamp providing the infrared part of the spectrum, as shown in Figure S19. Unless otherwise specified, a Fresnel lens was not used for these measurements. The schematic illustration of the experimental setup is shown in Fig. 5a. The PEC measurements were conducted following the protocols proposed by Ben-Naim et al. for III–V photoelectrodes[67]. LSV curves were scanned from a reverse bias (− 0.24 V vs. RHE) to more positive bias with a scan rate of 20 mV s$^{-1}$. The initial ~160 mV of each LSV scan was conducted without illumination to obtain the dark current. Measurements were halted when the current density approached 0 mA cm$^{-2}$ to prevent surface oxidation associated with passing anodic current[67].

Figure 5b presents the LSV curves of the platinized 3 J III–V photoelectrode at various operating pressures. Measurements were initially conducted at 1 bar, followed by 3 bar, 5 bar, and 8 bar, with a final measurement at 1 bar to confirm reproducibility. Notably, the LSV curves are nearly identical for all pressures, indicating that an increase in pressure has no significant impact on photocurrent in this configuration. This finding is unexpected as images taken during the measurements (Fig. 5d–g) clearly indicate variations in bubble formation on the photoelectrode at different pressures, with corresponding videos available in Supplementary Videos 5–8. In other words, the lack of photocurrent limitation suggests that the presence of bubbles does not hinder performance. To confirm this, we conducted measurements in the presence of 0.5 M sodium persulfate ($Na_2S_2O_8$) as an electron scavenger[68,69], which eliminates gas evolution (Fig. 5h and Supplementary Video 9). The resulting LSV curves (Fig. 5c) also show no pressure dependence, and the photocurrents are similar to those measured in the absence of electron scavenger (Fig. 5b), further supporting the conclusion that gas bubbles do not significantly impact photocurrent in this system.

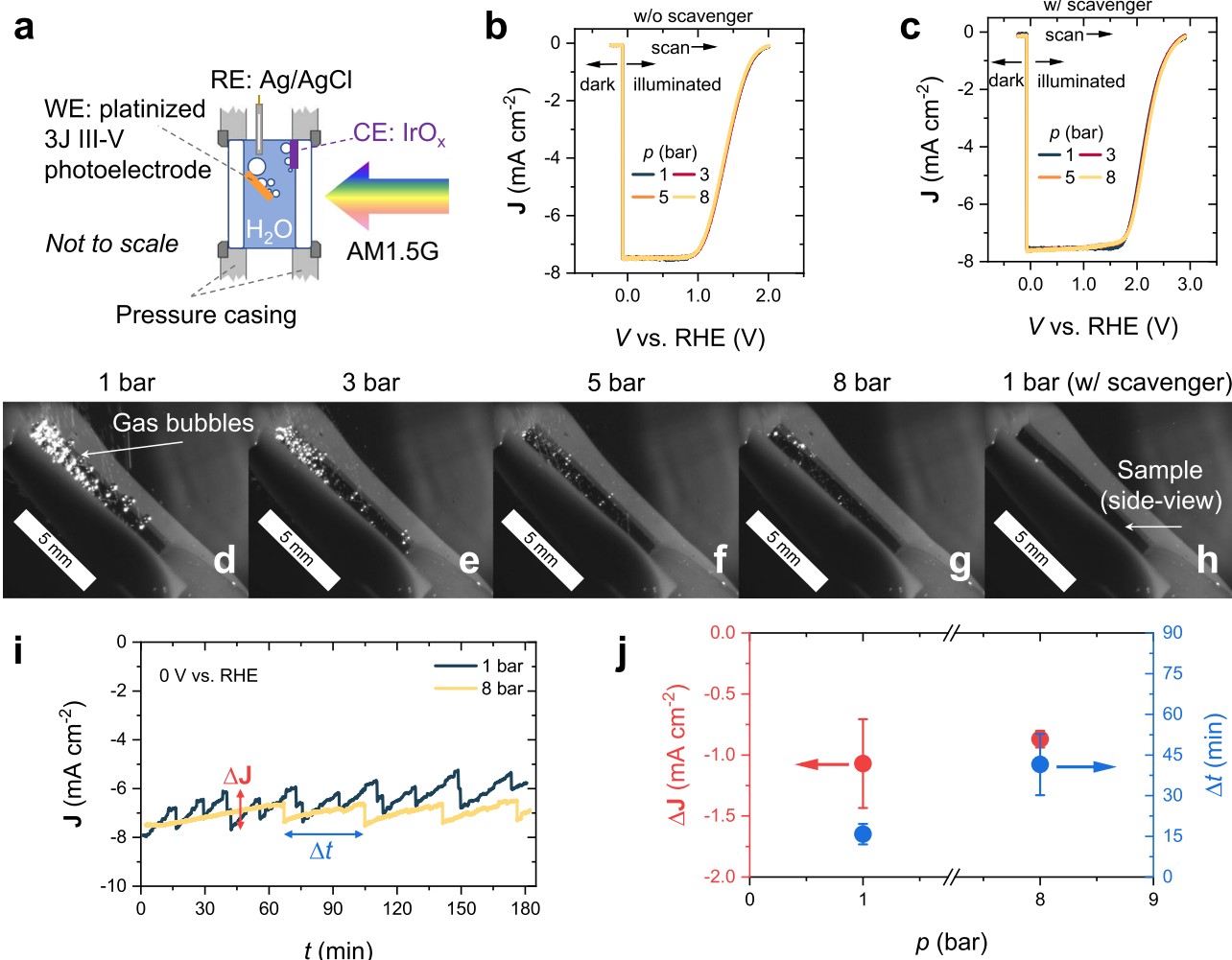

**Fig. 5 | Effect of pressure on the front-illuminated 3 J III-V based PEC water splitting cell. a** Schematic illustration of the cell configuration. The three-electrode configuration is constructed with the platinized 3 J III-V photoelectrode as the working electrode, an $IrO_x/TaO_x/Ti$ mesh serves as the counter electrode, and an Ag/AgCl reference electrode. The photoelectrode is tilted to ~45° from the horizontal plane to ensure a better camera angle for imaging. Linear sweep voltammetry (LSV) curves measured under AM1.5 G 1 sun illumination at different pressures (**b**) in a 0.1 M KP$_i$ solution (pH = 7 ± 0.1) and **c** in a 0.1 M KP$_i$ solution containing 0.5 M $Na_2S_2O_8$ as electron scavenger. The scan was performed from negative to positive applied bias at a scan rate of 20 mV s$^{-1}$, with the first ~160 mV measured in dark before turning on the illumination. No iR corrections was

conducted for the reported voltammograms. **d–g** Representative images of gas bubbles on the photoelectrode at 1–8 bar in a pH 7 ± 0.1, 0.1 M KP$_i$ solution. **h** Representative image of the photoelectrode during the reaction in a pH 7 ± 0.1, 0.1 M KP$_i$ solution containing 0.5 M $Na_2S_2O_8$. The scale bars in (**d–h**) represent 5 mm. **i** Chronoamperometry measurement at 0 V vs. RHE for 3 h at 1 bar vs. 8 bar in 0.1 M KP$_i$ solution (pH = 7 ± 0.1). **j** Change in photocurrent due to bubble detachment ($\triangle J$) and bubble growth time ($\triangle t$) at 1 bar vs. 8 bar. The operating pressure of the PEC water splitting cell is increased by supplying compressed $N_2$ gas. The electrolyte flow rate is kept at 4.6 mL s$^{-1}$. Error bars are the standard deviation of at least three measurements. The measurements were conducted at room temperature (~25 °C).

We also examined the behavior of bubbles over an extended operation period. We performed chronoamperometry at 1 bar and 8 bar, and the results are shown in Fig. 5i. To better describe the effect of bubble detachment to the photocurrent, we plot the $\Delta J$ and $\Delta t$ in the **J**-$t$ curves in Fig. 5j; $\Delta J$ represents the increase in photocurrent due to bubble detachment and $\Delta t$ is the bubble growth time prior to detachment. While bubbles form and detach cyclically (more frequently at 1 bar vs. 8 bar, i.e., $\Delta t_{1\,bar} < \Delta t_{8\,bar}$, as expected), their impact on photocurrent ($\Delta J$) remains similar (~10-15%), suggesting that bubbles do not persistently block catalytic sites.

The lack of pressure-dependence on the photocurrent means that bubbles do not reduce light absorption or block catalytically active surface sites on the photoelectrode. We speculate that three factors contribute to this observation. First, the platinized 3 J III-V photoelectrode exhibits high hydrophilicity (liquid contact angle ~35°, see Fig. S20). This means that bubbles detach more easily from the surface, minimizing coverage losses. Second, while bubbles can reflect and

scatter light, previous studies suggest that the smooth interface of gas bubbles leads to minimal diffuse scattering[29,33], with most reflected light being redirected away rather than lost. Additionally, due to refraction at the curved bubble surface, some of the redirected light can still be absorbed by other parts of the photoelectrode, rather than escaping entirely. Third, the Pt catalyst layer on the 3 J III-V cell is very thin (~1 nm) and likely forms dispersed nanoclusters rather than a continuous film, as illustrated in Fig. 6. Even when some clusters are temporarily blocked by bubbles, the long carrier diffusion lengths in III-V semiconductors[70,71] ensure that the photogenerated carriers can still reach Pt sites not covered by a bubble and thus contribute to the HER. Note that this is markedly different from metal oxide semiconductors like BiVO$_4$, which have diffusion lengths of 10–100 nm. Since this value is much shorter than the radius of a typical adhered gas bubble, nearly all the carriers generated directly below the gas bubble will recombine. Together, these factors explain why gas bubble formation in the platinized 3 J III-V photoelectrode does not significantly

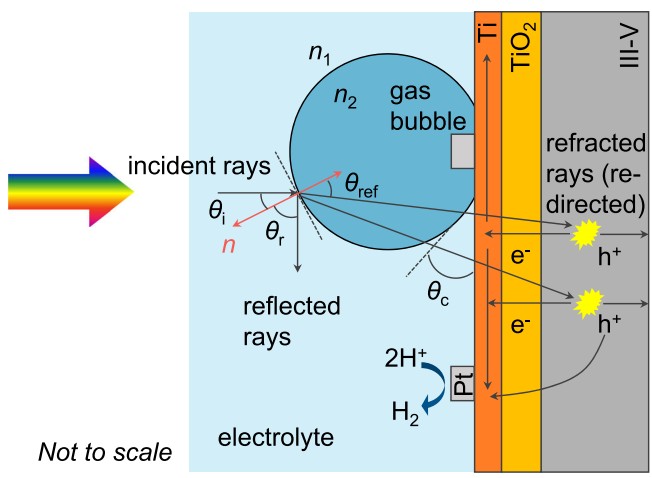

**Fig. 6 | Schematic illustration of the effect of gas bubble on the 3 J III-V photoelectrode.** $n_1$ and $n_2$ are the refractive index of the electrolyte and $H_2$ gas, respectively. $\theta_i$ denotes the incident angle of the light, $\theta_{ref}$ is the refractive angle, and $\theta_r$ is the reflection angle. $\theta_c$ is the contact angle of the gas bubble on the photoelectrode. Incident light can be reflected at the bubble/electrolyte interface (potentially lost) or refracted (redirected to other parts of the photoelectrode). The 3 J III-V PV cell was protected with Ti and $TiO_2$ layers, and the Pt catalyst (1 nm) was dispersed on the surface, where the hydrogen evolution reaction (HER) occurs. Some Pt sites are temporarily covered by bubbles and deactivated, but the long minority carrier lifetime and high carrier mobility in III-V semiconductors enables reactions at uncovered sites.

reduce the photocurrent, even at elevated pressure, enabling efficient hydrogen production at 8 bar without performance loss.

We briefly note that the impact of solar concentration on the front-illuminated integrated PEC water splitting cell configuration was also investigated. The LSVs and the normalized photocurrent ($J_{c=x}/J_{c=1}$) are shown in Fig. S21. Unlike the case of $BiVO_4$, the photocurrent increases proportionally with illumination intensity, i.e., the photocurrent at 3 suns is ~3-times that at 1 sun. This observation further confirms that light absorption and bubble-induced losses are not limiting the photocurrent in our platinized 3 J III-V photoelectrode.

Our findings suggest that the impact of gas bubble evolution on photocurrent is linked to the diffusion length of photoexcited charge carriers. Although light scattering by adhering bubbles does not significantly impact the photocurrent for photoelectrodes with long diffusion lengths, this is markedly different for short diffusion length materials, where most of the charge carriers generated directly below a surface-adsorbed bubble will recombine. In addition, for PEC cell designs where light must pass through bubble plume before reaching the photoelectrode, light scattering may still induce considerable optical losses[30,35,72]. Other important bubble-related effects, such as bubble-induced corrosion and localized photocorrosion due to bubble-induced light concentration[33], are beyond the scope of this study. Further investigation using advanced techniques like scanning photocurrent microscopy (SPCM)[31] is needed to fully assess these phenomena.

We will now discuss some general limitations of our study and provide an outlook for operating PEC devices at elevated pressure. First, in this initial demonstration of high-pressure PEC water splitting, only two representative and contrasting systems are selected: $BiVO_4$ as a metal oxide photoanode under back-illumination and a platinized triple-junction III-V photocathode under front-illumination. These photoelectrodes were intentionally chosen as benchmark photoelectrodes since they have been extensively studied in PEC water splitting research and beyond[16,17,66,73–77]. They are relatively stable and have demonstrated adequately high photocurrent; this ensures that performance variations in our study can be attributed primarily to

pressure-related effects rather than material-related factors, thus providing a reliable basis to assess their impact on device operation. At the same time, expanding this comparison to additional semiconductors, such as $\alpha$-$Fe_2O_3$ and $WO_3$ photoanodes or $Sb_2S_3$ and $Cu_2O$ photocathodes, would offer further insights into any material-specific pressure effects. These studies are part of our ongoing efforts and will be reported in due course.

On the system level, the benefits from operating PEC water splitting cells at elevated pressure should be ensured to outweigh the associated penalties. In addition to the suppression of bubble-induced losses, reducing or eliminating the need for downstream hydrogen compression represents a significant advantage. In practice, the elevated pressure PEC cell can be connected directly to high-pressure $H_2$ tanks, solid-state $H_2$ storage materials, or any applications that require elevated pressure operation (e.g., Haber-Bosch or hydrogenation reactors), as illustrated in Fig. S22. For instance, when compressing $H_2$ to 875 bar, doing so by starting from 1 bar consumes ~80% more energy than starting from 20 bar (~ 4.8 vs. ~2.7 kWh $kg^{-1}$)[78–80]. In principle, the system can be self-pressurized via gas evolution during operation, potentially avoiding additional energy input for pressurization other than the thermodynamic penalty of higher-pressure operation. In the pressure range of up to 20 bar, our recent analysis has shown that the benefits largely outweigh the thermodynamic penalty[35]. Nevertheless, we acknowledge that achieving and maintaining elevated pressure introduces technical and economic challenges, including the need for reinforced cell housings, thicker optical windows, enhanced sealing, pressure-rated tubing, and precise back-pressure control—all of which would increase capital and maintenance costs. A comprehensive techno-economic analysis is needed to evaluate the economic trade-offs and inform possible scale-up configurations.

Another factor not yet fully considered here is gas separation. We utilized precise fluid dynamic control of the electrolyte flow to ensure laminar condition and keep the evolved $O_2$ and $H_2$ gas bubbles closer to their respective electrodes, effectively minimizing gas crossover (see flow velocity colormaps in Fig. 2a). However, the purity of the separated gases has not been quantified. Such membraneless separation strategy via hydrodynamically guided flow has shown promise in simplifying system architecture[81,82], but we suspect that the incorporation of ion exchange membranes (e.g., Nafion® 117) might be required to achieve the highest purity and to guarantee safe operation. Moreover, the use of ion exchange membranes would enable the PEC cell to operate under differential pressure between the anodic and cathodic compartments. Prior studies have shown that hydrogen permeability through Nafion membranes remains largely independent of the applied pressure differential (up to 5 bar), indicating that pressure differences have a negligible effect on gas crossover[83].

In summary, we operated photoelectrochemical water splitting devices at elevated pressure up to 8 bar using two cell configurations. For the back-illuminated $BiVO_4$-based PEC cell, a saturation in photocurrent is observed under AM1.5 G illumination and 1 bar. An increase in operating pressure alleviates this saturation under high solar irradiance, improving the normalized photocurrent by ~40% at 5 bar under ~10× simulated sunlight, with marginal additional benefit at 8 bar. Direct operando imaging of the electrode surfaces reveals that this enhancement is primarily due to suppressed gas bubble evolution. In contrast, for the front-illuminated, integrated 3 J III-V PEC water splitting device, the photocurrent remains largely unaffected by gas bubble evolution and increased operating pressure. This can be attributed to the high hydrophilicity of the platinized top surface, but more importantly, to the long charge carrier diffusion length in III-V photoelectrodes. This causes the photocurrent to increase proportionally with light intensity up to a solar concentration factor of at least 3× for a platinized GaInP/GaAs/Ge PEC cell. Additionally, stability measurements revealed that operation at increased pressure does not significantly impact the degradation of these photoelectrode materials.

These results demonstrate the feasibility of operating PEC water splitting cells at elevated pressure, highlighting distinct effects of gas bubble evolution, pressure, cell configuration, and photoelectrode material properties. These findings offer fundamental insights that are directly relevant to PEC device design, bringing PEC water splitting a step closer to a practical application.

## Methods

### Design and fabrication of the high-pressure flow cell
The high-pressure flow cell (HPFC) was custom-designed and constructed in our workshop according to the schematics in Fig. 1b, with an exploded view of the cell design shown in Figure S1, and the digital photographs in Figure S2. The electrolyte was circulated using a high-pressure gear pump (LAB-9, GATHER Industrie). The bottom electrolyte reservoir has an approximate volume of ~400 mL, and a 3D printed structure made from transparent, high-temperature resistant plastic (VisiJet M2S-HT90) was installed to serve as a flow homogenizer / distributor. The flow homogenizer ensured that the electrolyte flow between the two electrodes in the reactor is laminar, as shown by the flow field obtained from particle image velocimetry (PIV) measurements (see Methods: Fluid flow field visualization and Fig. 2). The reactor part was 3D printed using stainless steel. A pair of sample holders can be installed, which allows a face-to-face arrangement of two 2 cm × 2 cm (maximum) samples with a distance of ~4 mm. The maximum effective area of the photoelectrodes is therefore ~4 cm$^2$, which is constrained by the size of the illumination window (see Figure S1). Note that the sample holders are exchangeable; in fact, we customized another pair of sample holders which enabled the front-illuminated photocathode to be tilted at 45° in the reactor. A pair of glass windows on the side walls were installed for better operando monitoring of the gas bubble evolution, see Figure S1. The cell was pressurized by supplying compressed $N_2$ (or $O_2$ for certain cases). The cell pressure was controlled using a back-pressure controller (Bronkhorst High-Tech, uncertainty: 0.2%), visible in Fig. 1b and Fig. S2. The highest pressure achieved is 8 bar(a), and no leakage or explosion happened during our experiments. A protective shield was constructed to isolate personnel from any potential safety risks.

We chose a membrane-free flow cell design to facilitate operando (or in situ) measurements at higher pressure, e.g., through particle image velocimetry (PIV) and multi-angle bubble imaging; without flow, the gas bubbles would accumulate and prohibit these experiments. However, during our experiments, we noticed that it is challenging to avoid mixing of the high-pressure gas with the liquid flow at the outlets. As a result, gas bubbles will form when the liquid re-enters the reactor due to cavitation and disturb the measurement (or lead to product crossover). Undesired cavitation happened more often at higher flow rates (e.g., >10 mL s$^{-1}$). We solved this problem by inserting capillary tubes (inner diameter of ~1.5 mm) underneath the electrolyte level and maintaining the electrolyte flow rate at a moderate level, i.e., ~4.6 mL s$^{-1}$. A schematic illustration of the liquid outlets as well as the gas inlets and outlets is shown in Fig. S23a, while a digital photograph of the capillary tube is presented in Fig. S23b. This solution was effective for most of our experiments, but we note that they might need to be evaluated on a case-to-case basis. For long-term operation of the high-pressure PEC device, a membrane or separator between the electrodes may be mandatory to assure effective product separation.

### Electrodeposition of the BiVO$_4$ photoelectrode
The BiVO$_4$ electrodeposition recipe used here follows that previously reported[74]. 3.32 g of potassium iodide (≥99%, Santa Cruz Biotechnology) was dissolved in 50 mL of deionized water (18.2 MΩ) to yield a 0.4 M potassium iodide solution. To this solution, 0.1 mL of nitric acid (>69.0%, Honeywell) and 0.04 M bismuth nitrate pentahydrate (≥98%, Acros Organics) were added. The mixture was magnetically stirred until all salts were completely dissolved. A 20 mL

ethanolic solution was prepared, which contains 0.225 M p-benzoquinone (≥98%, Alfa Aesar). The aqueous Bi-I precursor solution was then slowly added to the ethanolic solution under stirring, yielding a clear, dark red mixture.

BiOI nanosheet arrays were electrodeposited onto fluorine-doped tin oxide (FTO) substrates (3 × 3 cm$^2$, 3 mm thick, sheet resistance: 10 Ω sq$^{-1}$, Sigma Aldrich). The electrodeposition was carried out using a three-electrode configuration with a platinum coil (0.5 mm diameter) as the counter electrode and an Ag/AgCl (saturated KCl; XR300, Radiometer Analytical) electrode as the reference electrode. A constant potential of −0.1 V vs. Ag/AgCl was applied until a charge of 200 mC cm$^{-2}$ was reached, forming red-orange BiOI films.

The as-deposited BiOI films were subsequently coated with 50 μL cm$^{-2}$ of 0.2 M vanadyl acetylacetonate (≥99%, Acros Organics) solution in dimethyl sulfoxide (≥99.9%, VWR Life Science). The coated samples were annealed on a hot plate at 450 °C for 2 h with a ramping rate of 2 K min$^{-1}$ to induce the conversion to monoclinic BiVO$_4$. Residual $V_2O_5$ was removed by immersing the films in a 1 M NaOH (≥98%, Sigma Aldrich) solution for 30 min.

The morphology of the BiVO$_4$ films was characterized by scanning electron microscopy (SEM, GeminiSEM 360 instrument, Zeiss), see Fig. S7. The X-ray diffraction (XRD) pattern of the BiVO$_4$/FTO sample and the Tauc plot for indirect bandgap estimation are shown in Fig. S8a and S8b, respectively. The film thickness was determined to be ~10 μm using a DEKTAK 8 profilometer (Fig. S8c).

### Design and fabrication of the 3 J III-V photoelectrode
Platinized triple-junction (3 J) III-V photoelectrodes were fabricated following our previously reported procedure[84], The device architecture was based on a GaInP/GaAs/Ge (PV) cell (AZUR SPACE Solar Power GmbH) with a photoactive area of 1 cm$^2$. Each cell was supplied with a protective GaAs cap layer to prevent surface degradation during shipping and storage, and the back contact consisted of an Ag/Au metallization. Prior to the TiO$_2$ deposition, the GaAs cap layer was etched by wet chemical etching.

A 50 nm-thick TiO$_2$ protection layer was then deposited by atomic layer deposition (ALD) using a Picosun R200 Advanced system, with tetrakis(dimethylamido)titanium (TDMAT, 99%, Strem Chemicals, Inc.) and Milli-Q $H_2O$ serving as the precursor and oxidizer, respectively. The deposition temperature was maintained at 130 °C, resulting in an amorphous TiO$_2$ layer. To promote the hydrogen evolution reaction (HER), a 1 nm Pt catalyst layer was subsequently deposited by electron beam (e-beam) evaporation (base pressure of 2 × 10$^{-6}$ mbar) at a rate of 1 Å s$^{-1}$. For improved adhesion, a 2 nm Ti interlayer was deposited between the Pt and TiO$_2$ layers.

For photoelectrochemical (PEC) testing, the samples were mounted on the glass substrates using Ag conductive tape (3 M) applied along the substrate's midline. To ensure good ohmic contact, conductive Ag epoxy (Circuitworks CW2400, Chemtronics) was applied between the Ag tape and the back-side metallization of the III-V cell, followed by curing on a hot plate at 110 °C for 10 min. The electrode edges were sealed with a black two-part insulating silicone resin (101RF, Microset Products Ltd, see Figure S18), leaving only the active surface exposed. The electrochemically active area (~0.6–0.7 cm$^2$) was determined from digital images using ImageJ software (Fig. S18d). A representative photograph of the completed photoelectrode is shown in Fig. S18.

### PEC measurements
The instrumentation diagram of the test bench for the PEC measurement is shown in Fig. 1b. A dual light-source solar simulator (Wacom-WXS-100S-L2H AM 1.5GMM) was used as the light source. A set of Fresnel lenses (Sankuai, purchased from Amazon.de) with different focal lengths was used to increase the solar concentration. The solar irradiance was calibrated using a USB spectrometer (USB-2000+,

OceanOptics), see the Supplementary Note S1 and the measured irradiance shown in Fig. S3. The highest solar concentration obtained was ~10 suns for a Fresnel lens with a focal length of ~300 mm. We managed to illuminate the full sample area (~4 cm²) under ~10 suns, as shown by the photograph taken during the PEC tests in Fig. S4. However, it is worth noting that the irradiance was not homogenous throughout the sample (Fig. S3b and S3c). This inhomogeneity in solar irradiance is acceptable in our study, as our primary focus is pressure-dependence of the PEC water splitting cells performance. Nonetheless, this issue has to be dealt carefully for the real-world operation of such a PEC device under higher solar concentrations (e.g., $c > 10$). A photon homogenizer would be a viable solution, as reported in ref. [11].

For the PEC test with a back-illuminated $BiVO_4$ photoelectrode cell configuration, a small portion of the prepared $BiVO_4$ sample was etched using 1 M HCl (Fluka), on which electrical contact was made using a copper wire and conductive tape. The contact was then sealed properly with a black insulating two-part silicone resin (101RF, Microset). The $BiVO_4$ sample was then fixed into the sample holder (visible in Fig. S1) using a rubber O-ring and the silicon resin. The resulting photoactive area of the $BiVO_4$ samples was ~4 cm². A Pt mesh (0.198 mm diameter wire, 99.9%, Alfa Aesar) was glued onto a 30 mm × 30 mm × 3 mm (length × width × thickness) glass substrate and used as the counter electrode. After assembly, the working and counter electrode were arranged face-to-face, with a gap of ~4 mm. A miniature Ag/AgCl (saturated KCl) electrode (PalmSens BV) was used as the reference electrode. Note that the reference electrode was calibrated against a master RE (saturated KCl) prior to measurement, and potential shift ($\Delta V$) was ensured to be less than ±5 mV for each experiment. Electrochemical measurements were performed in a three-electrode configuration using a VersaSTAT 3 potentiostat/galvanostat (AMETEK). The measured potentials ($E_{Ag/AgCl}$) were converted to the reversible hydrogen electrode (RHE) scale using the Nernst equation:

$$E_{RHE}(V) = E_{Ag/AgCl}(V) + 0.0591 \times pH + E^0_{Ag/AgCl} \qquad (1)$$

here, $E_{RHE}$ is the applied potential vs. RHE and $E^0_{Ag/AgCl}$ is the potential of the reference electrode (0.198 $V_{RHE}$ at 25 °C). No iR corrections was conducted for all reported voltammograms.

For the front-illuminated PEC cell configuration, most of the setup (e.g., reference electrode, potentiostat) remained the same, except for the sample holders (see Fig. S1) and the counter electrode. The sample holders were changed to a self-made glass sample holder to enable irradiating the platinized 3 J III-V photoelectrode at a tilted angle (~45°, referring to the horizontal plane). An $IrO_x/TaO_x/Ti$ mesh was used as the counter electrode. To avoid the shadowing effect from the counter electrode and gas bubble evolution during the reaction, the counter electrode was placed above the light path, as shown by the schematic in Fig. 5a.

A 0.1 M potassium phosphate ($KP_i$, pH = 7 ± 0.1) solution was used as the electrolyte for most of the experiments and was prepared from $KH_2PO_4$ (Sigma-Aldrich, ≥99.0%) and $K_2HPO_4 \cdot 3H_2O$ (Sigma-Aldrich, ≥99.0%) to obtain the desired pH. Either 0.5 M $Na_2SO_3$ (Sigma-Aldrich, ≥99.0%) or 0.5 M $Na_2S_2O_8$ (Sigma-Aldrich, ≥99.0%) was added to the electrolyte as the hole or electron scavenger, respectively. The water used in all experiments was obtained from a Milli-Q integral system with a resistivity of 18.2 MΩ cm. Fresh electrolyte solution was prepared and used for each individual measurement, and the electrolyte was stored in a glass beaker.

The testing protocol for the back-illuminated $BiVO_4$ PEC cell was as follows: the electrolyte (0.1 M $KP_i$, pH = 7 ± 0.1) was purged with $O_2$ gas for at least 1 h, then transferred into the HPFC. The high-pressure $O_2$ gas supply was connected to the cell and maintained at the desired pressure. The rotary pump was started to circulate the electrolyte for about 15 min prior to the measurements. A 15 min gap was scheduled

to ensure the electrolyte flow was in steady state after adjusting the operating pressure. The solar concentration was varied by adding Fresnel lens with various focal lengths. A 5 min gap was scheduled before conducting the next experiment to minimize the effect of heating at different solar concentrations. A K-type thermocouple was used to monitor the electrolyte temperature at the outlets of the cell. All relevant parameters (e.g., pressure, electric potential, current, flow rate, etc.) were fed through the data acquisitions connected to a computer. The PEC tests of the front-illuminated 3 J III-V PEC cell followed a similar protocol, except that the compressed gas was changed to $N_2$. All the measurements were conducted at room temperature (~25 °C).

**Gas bubble visualization**

Two complementary imaging setups were used to observe gas bubble evolution on the electrodes from different angles during operation, as shown by the schematic in Fig. S6. For the back-illuminated PEC cell, a high-speed camera (HS5-Q Quad HD+ Resolution (2560 × 2048 pixels) with a Navitar Zoom 6000 Lens System, FasTec) was used to capture the side-view of gas bubble evolution, and the shadowgraphy of bubbles on the electrodes was captured using a LaVision camera system (Imager SX 6 M CCD camera, 2752 × 2200 pixels). For the front-illuminated PEC cell, only the high-speed camera was used to observe the side-view of gas bubbles. The camera was fixed higher than the 3 J III-V photoelectrode so that the surface of the sample was visible. Neutral density (ND) filters with various ND factors, purchased from Thorlabs, were used depending on the light condition to enhance the imaging quality. The localized telescopic imaging configuration achieved a spatial resolution of approximately 10 μm over a limited field of view (~5 × 4 mm), suitable for capturing detailed bubble dynamics in selected regions of interest (see Fig. 4e, f). The images were processed using ImageJ, with additional validation using DaVis 10 (LaVision). To enhance image clarity, we applied post-processing techniques in ImageJ, including background subtraction and contrast normalization. These helped improve visual distinction of gas bubbles, although lighting non-uniformities and optical limitations still affected the overall image quality. Nevertheless, since the experimental conditions are consistent across all tests, relative comparisons between our experimental results remain valid.

To enable full-field visualization of the bubble evolution on the $BiVO_4$ photoanode (2 cm × 2 cm), the setup was further modified as shown in the schematic illustration in Figure S14a. The same $BiVO_4$ and Ag/AgCl electrodes were kept as the working and reference electrodes, but the counter electrode was a platinum wire instead of the platinum mesh to reduce visual obstruction. The PEC cell was operated at 0.5 mA cm⁻² in static 0.1 M $KP_i$ (pH = 7 ± 0.1), and bubble distribution was recorded using a Zeiss Makro-Planar T* 2/100 lens coupled to a high-speed camera (HS5-Q, 2560 × 2048 pixels), with a resolution limit of ~45–50 μm in bubble diameter. The representative images of bubbles on the $BiVO_4$ surface under different pressures are shown in Fig. S14b.

**Fluid flow field visualization**

The electrolyte flow field between the two parallel electrodes was visualized using particle imaging velocimetry (PIV, LaVision). The working principle of the PIV measurement is schematically illustrated in Fig. S5a, and is briefly described as follows: polyamid spheres (20 μm diameter, LaVision) were seeded in liquid at a seeding density of approximately 1 million particles per mL, these microparticles have similar density as Milli-Q water. The mixture was stirred overnight to obtain a stable suspension. The seeded liquid was then fed into the flow cell and circulated using a rotary pump (LAB-9, GATHER Industrie). A double-pulse laser (532 nm) and light sheet optics were used to generate two consecutive laser sheets, and the illuminated particles were captured by a camera (LaVision Imager SX 6 M). The laser pulse

duration in our measurements ranged from 10 to 30 ms ($\Delta t$ in Fig. S5a), optimized based on the seeding density and the camera's spatial resolution. The digital photographs of the experimental setup are shown in Fig. S5b.

It is important to note that the fluid flow visualization results presented in Fig. 2 were obtained using the 'default' sample holders (as shown in Fig. S1). Changing the sample holders— such as when using the customized version for the front-illuminated PEC experiments— will alter the fluid flow field between the electrodes, even if the flow rate remains the same.

## Data availability
All data supporting the findings of this study are available within the main text and the Supplementary Information. Source data of the figures in the main text and Supplementary Information are provided with this paper. Source data are provided with this paper.

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

## Acknowledgements

We gratefully acknowledge the Helmholtz Association of German Research Centers (HGF) and the Federal Ministry of Education and Research (BMBF), Germany for supporting the development of solar powered technologies for $H_2$ generation within the framework of the Innovation Pool project "Solar $H_2$: Highly Pure and Compressed" and the Helmholtz Research Program "Materials and Technologies for the Energy Transition" (MTET). Part of the work was also carried out with the support of the Helmholtz Energy Materials Foundry (HEMF), a large-scale distributed research infrastructure founded by the German Helmholtz Association (GZ 714-48172-21/1), and the BMBF project "H2Demo" (No. 03SF0619A-K). We are grateful to AZUR SPACE Solar Power GmbH, who kindly provided us with the GaInP/GaAs/Ge triple-junction photovoltaic cells. We also acknowledge Christian Höhn, Lars Drescher, and Torsten Wagner for the construction of the pressurized water splitting cell, Dr. Babu Radhakrishnan and Yu-Lin Tsai for their assistance during experiments, and Dr. Peter Bogdanoff for the insightful discussions. F.F.A. also acknowledges support from CityUHK (project 9610621) and the Hong Kong Research Grant Council (RGC) under the ANR/RGC Joint Research Scheme (project A-CityU102/24). H.K. acknowledges support from the European Innovation Council (EIC) through the OHPERA project (grant agreement No. 101071010). F.L. also acknowledges support from Xi'an Jiaotong University under the 'Outstanding Young Researcher Program' (JX6JO050).

## Author contributions

Conceptualization, F.L., F.F.A., and R.v.d.K.; methodology, F.L., F.F.A., and R.v.d.K.; investigation, F.L., H.K., and D.S.B.; writing – original draft, F.L.; writing – review & editing, F.L., H.K., D.S.B., R.v.d.K., and F.F.A.; supervision, R.v.d.K. and F.F.A.; funding acquisition, F.L., R.v.d.K. and F.F.A.

## Competing interests

The authors declare no competing interests.
