## [Transparent Peer Review file · Nature Communications]

Photoelectrochemical water splitting cells at elevated pressure using BiVO₄ and platinized III-V semiconductor photoelectrodes

Corresponding Author: Dr Fatwa Abdi

Version 0:

Reviewer comments:

Reviewer #1

(Remarks to the Author)

The authors designed a high-pressure flow cell for PEC water splitting. A back-illuminated BiVO₄-based PEC cell and a front-illuminated platinized triple-junction (3J) III-V-based PEC cell were tested. It is found that at a solar concentration of 10, increasing the operating pressure from 1 to 5 bar causes the photocurrent to increase from 3 to almost 7 the photocurrent under 1 sun for the BiVO₄-based PEC cell. However, the photocurrent scales proportionally to the illumination intensity and does not saturate in the 3J III-V-based PEC cell. The findings demonstrate useful information for scaling up PEC water splitting cells. However, several significant comments and suggestions must be addressed before the manuscript can be recommended for publication:

1. Considering that the authors only tested a back-illuminated BiVO₄-based PEC cell and a front-illuminated platinized triple-junction (3J) III-V-based PEC cell in the manuscript, it is not clear whether "elevated pressure" is applicable to all photoelectrochemical water splitting cells. The title seems too big, and it is suggested the authors to modify the title to make it more focus.
2. The Abstract for Nature Communications is too long. According to the requirement of the journal, the abstract should be less than 150 words.
3. The authors demonstrated that elevated pressure can alleviate the effect of bubbles on the PEC performance of the BiVO₄-based PEC cell. Whether more energy should be input to maintain the elevated pressure? The energy conversion efficiency is important for possible scale-up applications. It is suggested the authors to discuss and comment on the total energy conversion efficiency in this elevated pressure system.
4. For possible scale-up applications, how to collect the produced hydrogen while maintain the elevated pressure in the system is important. It is suggested the authors to discuss and comments on this point.
5. This elevated pressure system is expected to produce a mixture of hydrogen and oxygen gases. How to effectively separate the hydrogen and oxygen gases is important for practical applications. It is suggested the authors to detail the possible methods or technologies for efficient separation of the produced hydrogen and oxygen gases.
6. It is important to confirm whether the PEC system can be stably operated at elevated pressure for a long time. Longer stability measurement to hundreds of hours is recommended to evaluate the reliability of the system.
7. This work aims to provide a new pathway for scaling PEC water splitting devices. It is suggested to discuss some recent works related to scale-up PEC devices to broaden the readership (Nat. Commun. 2025, 16, 990; Nat. Commun. 2025, 16, 2792; Nat. Energy 2024, 9, 272-284).

Reviewer #2

(Remarks to the Author)

This manuscript presents a novel and comprehensive study of the operation of photoelectrochemical (PEC) water splitting cells under high-pressure conditions. Using a custom-designed high-pressure laminar flow cell, the authors successfully demonstrate—for the first time—PEC hydrogen production at pressures up to 8 bar. The investigation of photoelectrodes such as BiVO₄ reveals how pressure-induced suppression of bubble formation influences photocurrent behavior. The experimental methodology is rigorous, and the findings are significant, offering practical guidance for the future scale-up of

PEC systems. However, several experimental aspects require clarification or further elaboration to make the manuscript more complete, particularly regarding system stability under prolonged operation.

1. It is recommended that the authors expand the scope of comparison, for instance by benchmarking the pressure-induced enhancement in photocurrent density against other established photoelectrode systems (e.g., $\alpha\text{-Fe}_2\text{O}_3$, WO_3 , Sb_2S_3).
2. Additional comparative data on bubble evolution under different pressure gradients should be included to systematically elucidate the effects of pressure on bubble nucleation, size, and distribution.
3. Although the manuscript focuses on short-term PEC performance enhancement, it is recommended to include stability data from long-term operation under pressurised conditions (e.g., 10 hours).
4. While the authors state that the bulk electrolyte temperature change is less than 2.5°C , local heating of the photoelectrode surface under 10 times solar irradiance over extended periods may affect interfacial dynamics, which should be carefully considered.
5. It would be valuable for the authors to briefly discuss how the high-pressure PEC module might be integrated into a practical solar water splitting system.
6. For better clarify the author's point of view and attract more readers, it is recommended to add citations. (Angew. Chem. Int. Ed. 2020, 59, 23094-23099; Nat. Sustain. 2025, doi.org/10.1038/s41893-025-01530-y; Matter 2024, 7, 2278-2293; Angew. Chem. Int. Ed. 2024, 136, e202317414.).

Reviewer #3

(Remarks to the Author)

This manuscript demonstrates the operating photoelectrochemical (PEC) water splitting cells at elevated pressures (up to 8 bar), addressing a bottleneck in scalable solar hydrogen production. The authors developed a custom high-pressure flow cell and investigated two configurations: (1) a back-illuminated BiVO_4 -based PEC cell, and (2) a front-illuminated integrated platinumized triple-junction (3J) III-V PEC device. For the BiVO_4 system, increased pressure suppressed gas bubble formation, resulting in up to a 50% improvement in photocurrent under 10 suns illumination. Operando imaging confirmed that the improved performance stemmed from reduced bubble coverage and size. In contrast, the 3J III-V PEC device exhibited pressure-independent photocurrent due to its long carrier diffusion length, thin dispersed Pt catalyst layer, and highly hydrophilic surface, which facilitated bubble detachment and minimized recombination losses. The manuscript underscores the configuration-dependent effects of pressure on PEC performance and highlights the benefit of high-pressure operation: directly generating pressurized hydrogen to reduce downstream compression energy costs. Overall, I think this is a great paper and the manuscript could meet the standard of Nature Communications after addressing the following issues.

1. The figure clarity should be improved. The authors need to make sure that the readers can gain the key breakthrough of this work by reading the introduction and figures. However, figures only contain the recorded data without highlighting the novelty of this work. For example, in Figure 1, they need to add Figure 1c to highlight the novelty as compared with the previous approach. Besides, there are too many labels in Figure 1b about the whole system, and I do not know how significance of this work as compared with other works. The same things happen in other figures, the authors need to improve the figure quality and highlight the key novelty.
2. To add pressure for hydrogen production, the long-term stability and safety issues require a second consideration. During large-scale applications, will the materials degrade under cyclic pressure/continuous pressure? Is there any auxiliary equipment required for pressure adding, and will this increase the total cost of the hydrogen production?
3. The authors focused more on the breakthrough at the systematic level, and the materials used are simple. However, I still want the authors to clarify why BiVO_4 is applied in this system. What is the key advantage as compared with other catalysts? Besides, instead of only simple materials characterization in supplementary information, is there any difference/breakthrough on the materials level?
4. Please expand the discussion on pressure-dependent bubble nucleation dynamics. The manuscript attributes performance gains to suppressed bubble evolution at higher pressure, but the underlying physical mechanism (e.g., Laplace pressure, critical bubble size, contact angle effects) should be further elaborated.
5. The observation of bubbles plays a significant role in understanding the reason for high water splitting. Therefore, I recommend improving the bubble imaging resolution. Can you specify the optical resolution limits of your cameras, and whether any contrast-enhancing techniques (e.g., DIC, phase contrast) were attempted? Suggest future tools (e.g., AFM, total internal reflection microscopy) to visualize nanobubbles.
6. Since this technique requires 10 sun for water splitting, the authors need to consider possible heat accumulation at high solar flux. Could longer operation at 10 suns lead to cumulative heating or catalyst degradation?

Reviewer #4

(Remarks to the Author)

Version 1:

Reviewer comments:

Reviewer #1

(Remarks to the Author)

The authors have addressed all comments from reviewers and the manuscript can be accepted for publication in the current version.

Reviewer #2

(Remarks to the Author)

The author has already provided excellent solutions to the previously raised questions, but there are a few minor points below that require clarification.

1. Could the author elaborate on how the maximum pressure value measured in the experimental setup was determined, and whether this value was constrained by safety considerations?
2. What methodology was employed to determine the effective electrode area used in the photoelectrochemical (PEC) tests?
3. How does variation in pressure influence the onset potential of the photoelectrode?

Reviewer #3

(Remarks to the Author)

I am happy with the revision and the manuscript could be accepted in its current form.

Reviewer #4

(Remarks to the Author)

Version 2:

Reviewer comments:

Reviewer #2

(Remarks to the Author)

The author has answered all my questions and I don't have any more for now.

Response to reviewers' comments

Manuscript ID: NCOMMS-25-32194-T

Title: Operating photoelectrochemical water splitting cells at elevated pressure

In this response letter, the reviewers' comments to our original manuscript are provided in **black**, and our point-by-point responses as well as the corresponding changes to the manuscript are shown in **blue**.

Reviewers' Comments:

Reviewer #1:

The authors designed a high-pressure flow cell for PEC water splitting. A back-illuminated BiVO₄-based PEC cell and a front-illuminated platinized triple-junction (3J) III-V-based PEC cell were tested. It is found that at a solar concentration of 10×, increasing the operating pressure from 1 to 5 bar causes the photocurrent to increase from 3× to almost 7× the photocurrent under 1 sun for the BiVO₄-based PEC cell. However, the photocurrent scales proportionally to the illumination intensity and does not saturate in the 3J III-V-based PEC cell. The findings demonstrate useful information for scaling up PEC water splitting cells. However, several significant comments and suggestions must be addressed before the manuscript can be recommended for publication:

1. Considering that the authors only tested a back-illuminated BiVO₄-based PEC cell and a front-illuminated platinized triple-junction (3J) III-V-based PEC cell in the manuscript, it is not clear whether “elevated pressure” is applicable to all photoelectrochemical water splitting cells. The title seems too big, and it is suggested the authors to modify the title to make it more focus.

Response: We thank the reviewer for the suggestion. Although it is likely that the “elevated pressure” condition and effects will apply to more than the photoelectrodes tested, we agree that it is more appropriate for our title to be more specific. The title is now changed from “Operating photoelectrochemical water splitting cells at elevated pressure” to “Operating photoelectrochemical water splitting cells at elevated pressure: a demonstration using BiVO₄ and platinized III-V semiconductor photoelectrodes”.

Associated changes to the Manuscript and Supplementary Information:

- *Title (Manuscript)*, page 1, line 1 – 2: “Operating photoelectrochemical water splitting cells at elevated pressure: a demonstration using BiVO₄ and platinized III-V semiconductor photoelectrodes”
 - *Title (Supplementary Information)*, page 1, line 1 – 3: “Operating photoelectrochemical water splitting cells at elevated pressure: a demonstration using BiVO₄ and platinized III-V semiconductor photoelectrodes”
2. The Abstract for Nature Communications is too long. According to the requirement of the journal, the abstract should be less than 150 words.

Response: We thank the reviewer for reminding us about this requirement. We have now shortened the Abstract to be 150 words.

Associated changes to the Manuscript:

- *Abstract (Manuscript), page 1 – 2, line 16 – 28* “Direct production of pressurized green hydrogen via photoelectrochemical (PEC) water splitting reduces the need for mechanical compression and mitigates bubble-related losses. However, existing demonstrations have been limited to atmospheric pressure. To bridge this gap, we designed, constructed, and tested a high-pressure flow cell for PEC water splitting using two configurations. In a back-illuminated BiVO₄-based PEC cell, increased pressure suppresses bubble evolution and alleviates photocurrent saturation under concentrated sunlight: at 10 suns, the photocurrent rises from 3× at 1 bar to ~7× at 5 bar. Direct operando imaging of the electrode surfaces confirms that this improvement comes primarily from suppressed bubble evolution. Conversely, a front-illuminated platinized triple-junction III-V-based PEC cell shows limited pressure dependence up to 8 bar due to its dispersed catalyst and long carrier diffusion length. These findings highlight the differing response of PEC devices to pressure and demonstrate a viable pathway toward scalable, high-pressure PEC hydrogen production.”

3. The authors demonstrated that elevated pressure can alleviate the effect of bubbles on the PEC performance of the BiVO₄-based PEC cell. Whether more energy should be input to maintain the elevated pressure? The energy conversion efficiency is important for possible scale-up applications. It is suggested the authors to discuss and comment on the total energy conversion efficiency in this elevated pressure system.

Response: We thank the reviewer for raising up this important point. Reducing the need for downstream hydrogen compression and suppression of bubble-induced losses represent significant benefits of operating PEC water splitting cells at elevated pressure. In our experiments, we supplied compressed gas to the cell for controlled conditions; however, in practical applications, the system can be self-pressurized through gas production during operation. At such, no additional energy input is required for system pressurization other than the thermodynamic penalty of higher-pressure operation. Our recent analysis shows that the benefits still largely outweigh this thermodynamic penalty in the pressure range of up to 20 bar.¹ Nevertheless, we fully acknowledge that increasing the operating pressure may also introduce nontrivial technical and economic challenges. These include the need for thicker glass components, more robust sealing strategies to prevent electrolyte and gas leakage, and the use of high-pressure-rated tubing and control systems—all of which can increase the capital and maintenance costs. Detailed techno-economic assessment beyond the scope of our current study is thus necessary to comprehensively evaluate the economic feasibility of operating PEC systems at elevated pressures. We have now added a brief discussion regarding the points above in the revised *Manuscript*.

Associated changes to the Manuscript:

- *Results (Manuscript)*, page 5, line 111 – 117: “High pressure operation was accomplished by supplying compressed gas into the cell while regulating the outflow rate of gases with a back-pressure controller. Note that this approach to regulate the pressure was chosen because it offers easy control and experimental convenience in a lab-based setting; in practical applications, PEC water splitting cells can self-pressurize through gas production during operation, requiring only a pressure relieve valve as a control mechanism.”
- *Results (Manuscript)*, page 29, line 537 – 555: “On the system level, the benefits from operating PEC water splitting cells at elevated pressure should be ensured to outweigh the associated penalties. In addition to the suppression of bubble-induced losses, reducing or eliminating the need for downstream hydrogen compression represents a significant advantage. In practice, the elevated pressure PEC cell can be connected directly to high-pressure H₂ tanks, solid-state H₂ storage materials, or any applications that require elevated pressure operation (e.g., Haber-Bosch or hydrogenation reactors), as illustrated in Fig. S22. For instance, when compressing H₂ to 875 bar, doing so by starting from 1 bar consumes ~80% more energy than starting from 20 bar (~4.8 vs. ~2.7 kWh kg⁻¹).⁷⁸⁻⁸⁰ In principle, the system can be self-pressurized via gas evolution during operation, potentially avoiding additional energy input for pressurization other than the thermodynamic penalty of higher-pressure operation. In the pressure range of up to 20 bar, our recent analysis has shown that the benefits largely outweigh the thermodynamic penalty.³⁵ Nevertheless, we acknowledge that achieving and maintaining elevated pressure introduces technical and economic challenges, including the need for reinforced cell housings, thicker optical windows, enhanced sealing, pressure-rated tubing, and precise back-pressure control—all of which would increase capital and maintenance costs. A comprehensive techno-economic analysis is needed to evaluate the economic trade-offs and inform possible scale-up configurations.”

4. For possible scale-up applications, how to collect the produced hydrogen while maintain the elevated pressure in the system is important. It is suggested the authors to discuss and comments on this point.

Response: We appreciate this valuable suggestion from the reviewer. We fully agree that gas separation and collection will be important in practical applications, but we emphasize that the primary focus of our current study is to explore the effect of pressure elevation on the performance of PEC water splitting cells. Nevertheless, we have now considered the possible solutions for collecting or utilizing the produced H₂ at high pressure in scale-up applications. Three possible strategies are proposed for storing or utilizing the high-pressure H₂ gas generated by the PEC water splitting cells, as illustrated schematically in **Fig. R1**. First, the produced H₂ can be stored in a high-pressure gas cylinders for transportation and later use. Second, it can be absorbed by the solid-state H₂ storage materials, such as MgNi, ZrCr, and LaNi₅-based metal hydrides. Third, the produced high-pressure H₂ can be directly fed into high-pressure hydrogen applications, such as Haber–Bosch reactors. This way, the elevated pressure can be maintained while collecting or directly utilizing the pressurized H₂.

We have now added a brief discussion in the revised *Manuscript* and *Supplementary Information*.

Figure R1. Schematic illustrations of three possible strategies for high-pressure H₂ storage/utilization. Option 1: the high-pressure H₂ is stored in a gas cylinder; Option 2: the high-pressure H₂ is absorbed by the solid-state H₂ storage materials, such as MgNi, ZrCr, and LaNi₅-based metal hydrides; Option 3: the high-pressure H₂ is directly fed into high-pressure hydrogen applications such as a Haber-Bosch reactor. Note that the oxygen is produced and released to the environment at nearly atmospheric pressure. Such a differential- pressure operation can be enabled by using a proton-exchange-membrane (e.g., Nafion® 117). Note that the modules such as gas-liquid separator, H₂ gas purification are not included in this schematic illustration.

Associated changes to the *Manuscript* and *Supplementary Information*:

- **Results (Manuscript), page 29, line 540 – 555:** “In practice, the elevated pressure PEC cell can be connected directly to high-pressure H₂ tanks, solid-state H₂ storage materials, or any applications that require elevated pressure operation (e.g., Haber-Bosch or hydrogenation reactors), as illustrated in Fig. S22. For instance, when compressing H₂ to 875 bar, doing so by starting from 1 bar consumes ~80% more energy than starting from 20 bar (~4.8 vs. ~2.7 kWh kg⁻¹).⁷⁸⁻⁸⁰ In principle, the system can be self-pressurized via gas evolution during operation, potentially avoiding additional energy input for pressurization other than the thermodynamic penalty of higher-pressure operation. In the pressure range of up to 20 bar, our recent analysis has shown that the benefits largely outweigh the thermodynamic penalty.³⁵ Nevertheless, we acknowledge that achieving and maintaining elevated pressure introduces technical and economic challenges, including the need for reinforced cell housings, thicker optical windows, enhanced sealing, pressure-rated tubing, and precise back-pressure

control—all of which would increase capital and maintenance costs. A comprehensive techno-economic analysis is needed to evaluate the economic trade-offs and inform possible scale-up configurations.”

- *Supplementary Information*, page 38: **Figure R1** has now been added as **Fig. S22** in the revised version.
5. This elevated pressure system is expected to produce a mixture of hydrogen and oxygen gases. How to effectively separate the hydrogen and oxygen gases is important for practical applications. It is suggested the authors to detail the possible methods or technologies for efficient separation of the produced hydrogen and oxygen gases.

Response: We appreciate this insightful and practical suggestion from the reviewer. Gas separation in the present work is achieved by carefully controlling the electrolyte flow dynamic, i.e., the produced H₂ and O₂ gas bubbles are kept close to the gas evolving electrodes by the laminar electrolyte flow (see the flow colormaps in **Fig. 2a**, *Manuscript*). Although the purity of the produced gases has not been quantified, this membraneless strategy has been shown to be promising.^{2,3} Nevertheless, it is suspected that incorporating ion-exchange membranes (e.g., Nafion® 117) would be required for extremely high purity and safe operation, especially under elevated pressure conditions. The use of ion-exchange membranes also allows operating the anodic and cathodic compartments of the PEC cells under differential pressure. Prior studies have shown that hydrogen permeability through Nafion membranes remains largely independent of the applied pressure differential (up to 5 bar), indicating that pressure differences have a negligible effect on gas crossover.⁴ We have now included this brief discussion in the revised *Manuscript*.

Associated changes to the *Manuscript*:

- *Results (Manuscript)*, page 30, line 556 – 568: “Another factor not yet fully considered here is gas separation. We utilized precise fluid dynamic control of the electrolyte flow to ensure laminar condition and keep the evolved O₂ and H₂ gas bubbles closer to their respective electrodes, effectively minimizing gas crossover (see flow velocity colormaps in Fig. 2a). However, the purity of the separated gases has not been quantified. Such membraneless separation strategy via hydrodynamically guided flow has shown promise in simplifying system architecture,^{81,82} but we suspect that the incorporation of ion exchange membranes (e.g., Nafion® 117) might be required to achieve the highest purity and to guarantee safe operation. Moreover, the use of ion exchange membranes would enable the PEC cell to operate under differential pressure between the anodic and cathodic compartments. Prior studies have shown that hydrogen permeability through Nafion membranes remains largely independent of the applied pressure differential (up to 5 bar), indicating that pressure differences have a negligible effect on gas crossover.⁸³”
6. It is important to confirm whether the PEC system can be stably operated at elevated pressure for a long time. Longer stability measurement to hundreds of hours is recommended to evaluate the reliability of the system.

Response: We agree with the reviewer that it is important to evaluate whether the elevated pressure has any impact on device stability. For the front-illuminated platinized III-V PEC cell, the photocurrent degradation remained comparable between 1 and 8 bar, as illustrated in **Fig. 5i** of the *Manuscript*. In the revised *Manuscript*, we have now included additional stability measurements for the back-illuminated, BiVO₄-based PEC water splitting cell under varying pressures. Chronoamperometry (CA) tests were conducted under potentiostatic conditions at 1.23 V vs. RHE, in a 0.1 M potassium phosphate buffer (KPi, pH = 7) with an electrolyte flow of 4.6 mL s⁻¹. Two illumination intensities were applied: 1 sun (AM1.5G) and 10 suns. Since uncatalyzed and undoped BiVO₄ samples were used in our study, photocurrent decrease is expected;^{5,6} however, we emphasize that the focus here is to understand if pressure elevation introduces any effect on degradation. The CA test at 1 sun was terminated when the photocurrent dropped by ~50%; while under 10 suns, the test was stopped at ~100% photocurrent loss. Interestingly, as shown in **Fig. R2**, no significant difference is observed in the normalized photocurrent plots at different pressure, both at 1 and 10 suns. This suggests that the BiVO₄ degradation mechanism is the same at different pressures. Overall, our study does not suggest any presence of pressure-induced degradation, but further detailed investigations are required to reveal the mechanisms. Finally, within the pressure range investigated, no gas or liquid leakage was detected during prolonged high-pressure operation, confirming the mechanical integrity and operational robustness of the system. This discussion has now been included in the *Manuscript* and the *Supplementary Information*.

Figure R2. Stability measurements of the back-illuminated, BiVO₄-based PEC water splitting cell under different pressures. (a) Chronoamperometry (CA) measurements under AM1.5G 1 sun illumination; (b) the same CA data divided by the respective J_{\max} for clearer comparison. (c) and (d) present the corresponding CA results under concentrated illumination (10 suns). In all tests, undoped and uncatalyzed BiVO₄ was used as the photoanode, with a platinum mesh as the counter electrode (CE), and Ag/AgCl as the reference electrode. The PEC cells were operated potentiostatically at 1.23 V vs. RHE in a 0.1 M potassium phosphate (KP_i) buffer solution (pH = 7), an electrolyte flow rate of 4.6 mL s⁻¹ was maintained during the stability measurements. The CA test was terminated upon ~50% photocurrent loss at 1 sun and near-complete degradation at 10 suns.

Associated changes to the Manuscript and Supplementary Information:

- **Results (Manuscript), page 21, line 362 – 371:** “To investigate whether pressure elevation introduces any effect on the stability of the back-illuminated BiVO₄ photoanode, potentiostatic experiments were performed with the same PEC cell at 1.23 V vs. RHE. Illumination intensities of 1 sun and 10 suns were applied. As shown in Fig. S15, the photocurrent decreases with time, indicating degradation; this is expected since our BiVO₄ is uncatalyzed and has no protection layer.^{25,63} However, the photocurrent decrease remains similar at different pressures, suggesting that pressure elevation has minimal impact on the degradation behavior of the BiVO₄ photoelectrode. Moreover, no gas or liquid leakage was observed during the extended high-pressure operation, indicating reliable structural mechanical integrity and operational stability of the system.”

- *Supplementary Information*, page 25: **Figure R2** has now been added as **Fig. S15** in the revised version.
7. This work aims to provide a new pathway for scaling PEC water splitting devices. It is suggested to discuss some recent works related to scale-up PEC devices to broaden the readership (Nat. Commun. 2025, 16, 990; Nat. Commun. 2025, 16, 2792; Nat. Energy 2024, 9, 272-284).

Response: We thank the reviewer for this valuable suggestion and for pointing out these important relevant works. We have now expanded the discussion in the revised *Manuscript* to cover more recent advances in the field, and the suggested references have been included as Refs. 6, 20, and 21.

Associated changes to the *Manuscript*:

- *Introduction (Manuscript)*, page 2, line 38 – 46: “For PC water splitting systems, near-unity conversion yields have been achieved under UV irradiation.³ Moreover, the feasibility of scale-up to larger areas using photocatalyst sheets was demonstrated with a 100 m² outdoor prototype panel reactor system,³⁻⁵ although the maximum solar-to-hydrogen (STH) efficiency was only ~0.76%.⁵ A slightly higher STH efficiency of 1.21% was recently demonstrated with a PC water splitting system that utilizes an I₃⁻/I⁻ redox mediator. Employing MoSe₂-loaded halide perovskites (CH(NH₂)₂PbBr_{3-x}I_x) for H₂ evolution and NiFe-layered double hydroxide-modified BiVO₄ for O₂ evolution, this 700 cm² system demonstrated stable operation over one week under natural sunlight.⁶”
- *Introduction (Manuscript)*, page 3, line 59 – 62: “With record-high STH efficiencies demonstrated, efforts are now focused on scaling up PEC water splitting systems. In general, PEC water splitting can be scaled by increasing the photoactive area per device,²⁰ by deploying a greater number of devices,^{2,20,21} and/or by utilizing higher solar concentrations.^{5,9,22}”

Reviewer #2:

This manuscript presents a novel and comprehensive study of the operation of photoelectrochemical (PEC) water splitting cells under high-pressure conditions. Using a custom-designed high-pressure laminar flow cell, the authors successfully demonstrate—for the first time—PEC hydrogen production at pressures up to 8 bar. The investigation of photoelectrodes such as BiVO_4 reveals how pressure-induced suppression of bubble formation influences photocurrent behavior. The experimental methodology is rigorous, and the findings are significant, offering practical guidance for the future scale-up of PEC systems. However, several experimental aspects require clarification or further elaboration to make the manuscript more complete, particularly regarding system stability under prolonged operation.

Response: We sincerely appreciate the reviewer's positive feedback. We have carefully addressed all the listed comments/suggestions and believe that the revision has significantly strengthened our study.

The point-to-point responses are as follow:

1. It is recommended that the authors expand the scope of comparison, for instance by benchmarking the pressure-induced enhancement in photocurrent density against other established photoelectrode systems (e.g., $\alpha\text{-Fe}_2\text{O}_3$, WO_3 , Sb_2S_3).

Response: We appreciate the reviewer's suggestion. While we agree that extending the scope of comparison to other semiconductors such as $\alpha\text{-Fe}_2\text{O}_3$, WO_3 , Sb_2S_3 , and Cu_2O would further enrich the understanding of pressure effects across different material platforms, we emphasize that the focus of our study is to demonstrate the first high-pressure PEC water splitting cells. Therefore, we intentionally selected two representative and contrasting systems: a metal oxide (BiVO_4) and a platinized triple-junction III-V semiconductor. This selection encompasses both photoanode and photocathode configurations as well as front- and back-illumination cell geometries. We believe this selection provides sufficient breadth to establish the general feasibility and performance trends under elevated pressure. We therefore kindly ask the reviewer's understanding that further expansion of scope to include other promising semiconductors is part of our planned future work and will be reported in due course. We have added a brief discussion to acknowledge this point in the revised *Manuscript*. Moreover, as also noted in our response to Comment 1 of Reviewer #1, we have modified the *Manuscript* title to more accurately reflect the scope of the current study.

Associated changes to the *Manuscript* and *Supplementary Information*:

- *Results (Manuscript)*, page 28 – 29, line 521 – 536: “We will now discuss some general limitations of our study and provide an outlook for operating PEC devices at elevated pressure. First, in this initial demonstration of high-pressure PEC water splitting, only two representative and contrasting systems are selected: BiVO_4 as a metal oxide photoanode under back-illumination and a platinized triple-junction III-V photocathode under front-illumination. These photoelectrodes were intentionally chosen as benchmark photoelectrodes since they have been

extensively studied in PEC water splitting research and beyond.^{16,17,66,73-77} They are relatively stable and have demonstrated adequately high photocurrent; this ensures that performance variations in our study can be attributed primarily to pressure-related effects rather than material-related factors, thus providing a reliable basis to assess their impact on device operation. At the same time, expanding this comparison to additional semiconductors, such as α - Fe_2O_3 and WO_3 photoanodes or Sb_2S_3 and Cu_2O photocathodes, would offer further insights into any material-specific pressure effects. These studies are part of our ongoing efforts and will be reported in due course.”

- *Title (Manuscript), page 1, line 1 – 2:* “Operating photoelectrochemical water splitting cells at elevated pressure: a demonstration using BiVO_4 and platinized III-V semiconductor photoelectrodes”
 - *Title (Supplementary Information), page 1, line 1 – 3:* “Operating photoelectrochemical water splitting cells at elevated pressure: a demonstration using BiVO_4 and platinized III-V semiconductor photoelectrodes”
2. Additional comparative data on bubble evolution under different pressure gradients should be included to systematically elucidate the effects of pressure on bubble nucleation, size, and distribution.

Response: We appreciate the reviewer’s insightful suggestion. To further elucidate the pressure effects on gas bubble evolution, we have performed additional experiments focusing on the oxygen bubble dynamics from the surface of the BiVO_4 photoanode. To enhance visualization over the full $2\text{ cm} \times 2\text{ cm}$ electrode area, the experimental setup was modified, as shown in **Fig. R3a**. In short, the same BiVO_4 and Ag/AgCl electrodes were kept as the working and reference electrodes, but the counter electrode was a platinum wire instead of the platinum mesh to reduce visual obstruction. The PEC cell was operated galvanostatically at 0.5 mA cm^{-2} in 0.1 M KP_i ($\text{pH} = 7$). A macroscopic Zeiss Makro-Planar T* 2/100 lens coupled with a high-speed camera (HS5-Q, 2560×2048 pixels) was used to capture bubble distribution, achieving a detection threshold of $\sim 45\text{--}50\text{ }\mu\text{m}$ in bubble diameter.

Representative images (**Fig. R3b**) clearly show that increasing pressure significantly suppresses gas bubble formation on the BiVO_4 surface, consistent with localized observations presented in **Fig. 4e–f** of the *Manuscript*. At 1 bar, bubbles were randomly distributed across the porous BiVO_4 layer, with larger bubble sizes observed near the top of the electrode—which can be attributed to hydrostatic pressure gradients. However, bubble presence was already negligible at pressures above 2 bar. This insight should be considered for the scale-up design of PEC systems.

We also appreciate the reviewer’s suggestion to investigate bubble nucleation events. Unfortunately, direct observation of nucleation events was not feasible due to the limited resolution of our current imaging setup. However, related studies, such as a recent work on bubble evolution on TiO_2 micro-photoelectrodes,⁷ confirm that elevated pressure facilitates bubble detachment due to increased dissolved gas saturation—aligned with previous simulation results.⁸ Finally, we emphasize that the topic of gas bubble dynamics attracts increasing interests from researchers in the

field, and a full understanding of this complex dynamics under pressure remains an open question. Advanced experimental and theoretical techniques such as atomic force microscopy⁹⁻¹¹ and molecular dynamics simulations^{12,13} will be essential for future investigation.

Additional discussion has now been added to the revised *Manuscript* and *Supplementary Information*.

Figure R3. Macroscopic shadowgraphy of oxygen bubble evolution from the BiVO₄ photoanode. (a) Schematic of the measurement setup. (b) Representative images of bubble formation at 1, 2, and 3 bar, respectively. The working electrode (BiVO₄, 2 cm × 2 cm active area) is identical to that used in the main study. A platinum wire was employed as the counter electrode (instead of the platinum mesh) to minimize visual obstruction, and a saturated Ag/AgCl electrode served as the reference. The PEC cell was operated galvanostatically at 0.5 mA cm⁻² in 0.1 M potassium phosphate buffer (KPi, pH = 7), with the electrolyte held static. AM1.5G 1 sun illumination was used. Imaging was conducted using a Zeiss Makro-Planar T* 2/100 lens coupled to a high-

speed camera (HS5-Q, 2560 × 2048 pixels), yielding a bubble detection threshold of approximately 45–50 μm. Note that gas bubbles observed at the top of the frames correspond to the region of electrical contact, where a small amount of silicone epoxy was applied for insulation.

Associated changes to the *Manuscript* and *Supplementary Information*:

- *Results (Manuscript)*, page 20, line 352 – 361: “To further elucidate the effects of pressure on gas bubble evolution throughout the whole surface of the 2 cm × 2 cm BiVO₄ photoelectrode, we conducted macroscopic imaging of oxygen bubbles. The experimental setup was modified accordingly (Fig. S14a) and introduced in the *Methods* section. At 1 bar, bubbles appeared randomly across the porous BiVO₄ surface (Fig. S14b), with larger bubble sizes observed near the top of the electrode; we tentatively attribute this observation to the presence of hydrostatic pressure gradients. This insight is useful for the scale-up of PEC devices, as non-uniform bubble detachment may result in spatial variations in performance. However, at pressures above 2 bar, bubble presence was already negligible (Fig. S14b). This observation is consistent with our localized observations shown in Fig. 4 e-f.”
- *Results (Manuscript)*, page 17, line 315 – 333: “Although bubble nucleation events could not be directly resolved, recent work on bubble evolution on TiO₂ micro-photoelectrodes showed that elevated pressure facilitates bubble detachment, primarily due to higher dissolved gas saturation.⁴⁸ Consistent with these findings, our measurements (Fig. 4e–f) revealed smaller and fewer bubbles under elevated pressure. This behavior can be tentatively explained by the formation of a supersaturated boundary layer (SSBL) of dissolved gas near the (photo)electrode at higher pressure. Our recent simulation showed that the SSBL facilitates bubble detachment from the surface of the (photo)electrode via Marangoni convection.⁴⁹ Additional contributing factors may include an increased Laplace pressure and a larger critical radius for nucleation at higher pressures, both of which inhibit initial bubble formation. Interfacial properties, such as the gas–liquid contact angle, may also play a role in pressure-dependent detachment dynamics, but direct experimental measurement of contact angles and nano/micro bubbles on semiconductor surfaces (e.g., BiVO₄ photoelectrode) under elevated pressure remains technically challenging. High-resolution tools such as atomic force microscopy and spectroscopy techniques^{50–60} and molecular dynamics simulation^{61,62} will be essential, which are beyond the scope of this study. Furthermore, the porous morphology of our BiVO₄ samples (see Fig. S6) introduces additional complexity, as it may lead to random spatial distribution of nanobubbles, making their detection even more complicated.”
- *Methods (Manuscript)*, page 38 – 39, line 757 – 766: “To enable full-field visualization of the bubble evolution on the BiVO₄ photoanode (2 cm × 2 cm), the setup was further modified as shown in the schematic illustration in Fig. S14a. The same BiVO₄ and Ag/AgCl electrodes were kept as the working and reference electrodes, but the counter electrode was a platinum wire instead of the platinum mesh to reduce visual obstruction. The PEC cell was operated at 0.5 mA cm⁻² in static 0.1 M KP_i (pH = 7), and bubble distribution was recorded using a Zeiss Makro-Planar T* 2/100 lens coupled to a high-speed camera (HS5-Q, 2560 × 2048

pixels), with a resolution limit of $\sim 45\text{--}50\ \mu\text{m}$ in bubble diameter. The representative images of bubbles on the BiVO_4 surface under different pressures are shown in Fig. S14b.”

- *Supplementary Information*, page 23 – 24: **Figure R3** has now been added as **Fig. S14** in the revised version.
3. Although the manuscript focuses on short-term PEC performance enhancement, it is recommended to include stability data from long-term operation under pressurised conditions (e.g., 10 hours).

Response: We thank the reviewer for this constructive suggestion. As noted in our response to Comment 6 of Reviewer #1, additional long-term stability measurements have now been conducted on the back-illuminated BiVO_4 -based PEC water splitting cell under different pressures. Chronoamperometry (CA) tests were conducted under potentiostatic conditions at 1.23 V vs. RHE, in a 0.1 M potassium phosphate buffer (KPi , $\text{pH} = 7$) with an electrolyte flow of $4.6\ \text{mL s}^{-1}$. Two illumination intensities were applied: 1 sun (AM1.5G) and 10 suns. Since uncatalyzed and undoped BiVO_4 samples were used in our study, photocurrent decrease is expected;^{5,6} however, we emphasize that the focus here is to understand if pressure elevation introduces any effect on degradation. The CA test at 1 sun was terminated when the photocurrent dropped by $\sim 50\%$; while under 10 suns, the test was stopped at $\sim 100\%$ photocurrent loss. Interestingly, as shown in **Fig. R2**, no significant difference is observed in the normalized photocurrent plots at different pressure, both at 1 and 10 suns. This suggests that the BiVO_4 degradation mechanism is the same at different pressures. Overall, our study does not suggest any presence of pressure-induced degradation, but further detailed investigations are required to reveal the mechanisms. Finally, within the pressure range investigated, no gas or liquid leakage was detected during prolonged high-pressure operation, confirming the mechanical integrity and operational robustness of the system. This discussion has now been included in the *Manuscript* and the *Supplementary Information*.

Associated changes to the *Manuscript* and *Supplementary Information*:

- *Results (Manuscript)*, page 21, line 362 – 371: “To investigate whether pressure elevation introduces any effect on the stability of the back-illuminated BiVO_4 photoanode, potentiostatic experiments were performed with the same PEC cell at 1.23 V vs. RHE. Illumination intensities of 1 sun and 10 suns were applied. As shown in Fig. S15, the photocurrent decreases with time, indicating degradation; this is expected since our BiVO_4 is uncatalyzed and has no protection layer.^{25,63} However, the photocurrent decrease remains similar at different pressures, suggesting that pressure elevation has minimal impact on the degradation behavior of the BiVO_4 photoelectrode. Moreover, no gas or liquid leakage was observed during the extended high-pressure operation, indicating reliable structural mechanical integrity and operational stability of the system.”
- *Supplementary Information*, page 25: **Figure R2** has now been added as **Fig. S15** in the revised version.

4. While the authors state that the bulk electrolyte temperature change is less than 2.5°C, local heating of the photoelectrode surface under 10 times solar irradiance over extended periods may affect interfacial dynamics, which should be carefully considered.

Response: We are grateful to the reviewer for highlighting this important issue. To better assess the thermal effects, we have now measured the bulk electrolyte temperature using a K-type thermocouple during stability tests under 10 suns illumination (see setup in **Fig. S10a**, *Supplementary Information*). The results, shown as symbols in **Fig. R5a**, indicate that the electrolyte temperature increased by approximately 5 °C at 1 bar and 3 °C at 5 bar under continuous illumination. Notably, the similar temperature trends observed at different pressures suggest that our performance comparisons under elevated pressures remain valid.

Nevertheless, we fully agree with the reviewer that bulk electrolyte temperature measurements do not necessarily reflect localized heating at the photoelectrode surface, which may influence interfacial reaction kinetics and material stability under concentrated light.¹⁴⁻¹⁶ Direct measurement of the photoelectrode surface temperature is challenging with our high-pressure PEC cell geometry, so we simulated the temperature distribution at the photoanode surface using a heat transfer model, detailed below.

Model descriptions

Problem definition and assumptions

We simplified the PEC cell geometry (see **Fig. 1b**, *Manuscript* and **Fig. S1**, *Supplementary Information*) for simulation of coupled heat transfer in solids and fluids. The BiVO₄/FTO photoanode was modelled as a quartz substrate with equivalent thickness. The fluid domain (electrolyte) was assumed to be liquid water under fully developed laminar flow, with inlet velocity taken from PIV measurements (**Fig. 2**, *Manuscript*). An inward heat flux of 5600 W m⁻² (corresponding to 56% absorption under 10 suns, per Ref. ¹⁷) was applied to the illuminated quartz surface. The edges of the electrode, which are in contact with the rubber gasket and stainless-steel cell body in our setup (see **Fig. S1**, *Supplementary Information*), as well as the right-side boundary of the fluid domain (in contact with the counter electrode), were treated as adiabatic boundaries, as shown in **Fig. R4a** and **R4c**. As a result, the simulated surface temperature of the photoelectrode represents a worst-case scenario for thermal accumulation.

Fluid dynamics

The following continuity equation for incompressible Newtonian fluids that conserves mass was considered,

$$\rho \nabla \cdot \mathbf{v} = 0 \quad (\text{R1})$$

where \mathbf{v} is the velocity field. The density of the fluid phase (ρ) is constant. This means that the only way to change the mass of the computational domain (Ω) is by convection, which is expressed in the following equation.

$$\rho(\mathbf{v} \cdot \nabla) \mathbf{v} = \nabla \cdot [-p\mathbf{I} + \mathbf{K}] + \mathbf{F} \quad (\text{R2})$$

$$\mathbf{K} = \mu(\nabla \mathbf{v} + (\nabla \mathbf{v})^T) \quad (\text{R3})$$

Momentum conservation was solved using the Navier-Stokes equation in (eqn. R2). p is pressure, \mathbf{I} is the identity tensor, while \mathbf{F} is the external volume forces, e.g., gravity. \mathbf{K} is the viscous stress tensor, and expressed in (eqn. R3), while μ is dynamics viscosity.

Boundary conditions

The liquid inlet was considered as fully-developed laminar flow,

$$\mathbf{v} = \left(0, 1.5U \left\{ 1 - \left(\frac{2x}{L_x} \right)^2 \right\} \right) \quad (\text{R4})$$

where L_x is the width of the flow channel and equals 4 mm, and U denotes the average inlet flow velocity. The U values were obtained from our PIV measurement presented in **Fig. 2a** (*Manuscript*) and set as 7.33 mm s⁻¹. The outlet pressure was set as 0 bar and 4 bar, respectively, to mimic the pressure settings of 1 bar and 5 bar in our experiments. The side walls were treated as no slip boundaries, as shown in **Fig. R4b**.

Heat transfer in solids and fluids

The heat transfer in the solid part was solved using the following equation.

$$d_z \rho C_p \mathbf{v} \cdot \nabla T + \nabla \cdot \mathbf{q} = d_z Q + q_0 \quad (\text{R5})$$

$$\mathbf{q} = -d_z k \nabla T \quad (\text{R6})$$

Here, d_z is the thickness of the domain and equals 30 mm. C_p is the specific heat capacity of the fluid, $\mathbf{v} \cdot \nabla T$ represents the convection term, and $\nabla \cdot \mathbf{q}$ solves heat conduction (divergence of heat flux). The heat flux \mathbf{q} is defined in (eqn. R6), with k representing the thermal conductivity of the material ($k = 1.4 \text{ W (m K)}^{-1}$ for quartz glass). On the right-hand side of the equation, $d_z Q$ denotes the volumetric heat source, and q_0 is the surface heat source.

For the fluid domain, the same equations were used, with nonisothermal flow assumed. The viscous dissipation term was included as

$$Q_{\text{vd}} = \tau : \nabla \mathbf{v} \quad (\text{R7})$$

where Q_{vd} is the dissipative heat generation term, τ is the viscous stress tensor.

Boundary conditions

An inward heat flux of 5600 W m⁻² (corresponding to 56% absorption under 10 suns, per Ref. ¹⁷) was applied at the illuminated section of the BiVO₄ photoanode. The edges of the electrode, which are in contact with the rubber gasket and stainless-steel cell body in our setup (see **Fig. S1**, *Supplementary Information*), as well as the right-side boundary of the fluid domain (in contact with the counter electrode), were treated as adiabatic boundaries, as shown in **Fig. R4c**. The numerical settings for other boundaries are shown in **Fig. R4c**.

Numerical treatment

The model was solved in fully coupled mode using COMSOL Multiphysics® 6.2 with the PARDISO solver. A P1+P1 scheme was used for pressure-velocity coupling, and time integration was performed using the generalized alpha method with adaptive time stepping. The computational domain was discretised into 79,927 elements, yielding an average mesh quality of 0.86 (with 1.0 indicating an ideal mesh).

Simulations were run on a high-performance workstation (Intel Xeon E5-2650 v2, 64 cores, 256 GB RAM) with a relative tolerance of 0.005.

Figure R4. Two-dimensional heat transfer model used for our simulations. The simplified geometry of our setup is shown in (a). (b) Governing equations and boundary conditions for the fluid flow model. (c) Heat transfer model in the solid and fluid domains. The BiVO_4/FTO sample was modelled using a quartz glass piece, with the electrolyte was treated as liquid water. An inward heat flux of 5600 W m^{-2} (corresponding to 56% absorption under 10 suns, per Ref. ¹⁷) was applied at the illuminated section of the BiVO_4 photoanode. The edges of the electrode, which are in contact with the rubber gasket and stainless-steel cell body in our setup (see **Fig. S1, Supplementary Information**), as well as the right-side boundary of the fluid domain (in contact with the counter electrode), were treated as adiabatic boundaries in panels (a) and (c). As a result, the simulated surface temperature of the photoelectrode represents a worst-case scenario for thermal accumulation. The liquid flow is

considered as the fully-developed laminar flow, with the average velocity obtained from our PIV measurements (shown in **Fig. 2a**, *Manuscript*).

Figure R5a compares the simulated surface temperature with experimental bulk measurements. Both show ~ 5 °C rise under steady-state illumination, validating the model's ability to capture thermal behavior. In addition, **Figure R5b-R5d** shows a maximum surface temperature rise of ~ 13 °C at the photoelectrode under both pressure conditions. While this estimate may be somewhat exaggerated due to simplified boundary assumptions—specifically, the adiabatic treatment of the electrode sealing interfaces and the electrolyte boundary adjacent to the counter electrode—it nonetheless underscores the potential for localized heating at the photoelectrode surface. Such a temperature increase can indeed facilitate reaction kinetic¹⁸ and accelerate photoelectrode material degradation¹⁴. These findings emphasize the importance of thermal management in high-pressure PEC systems under concentrated sunlight. Advanced cooling strategies, such as those demonstrated in Ref. ¹⁹ and Ref. ^{6,17}, can be implemented to prevent photoelectrode overheating under concentrated sunlight.

Importantly, the similar thermal trends observed in both our measurements and simulations at different pressures support the validity of our comparative performance analysis, as illustrated in **Fig. R5**. We have added a new *Supplementary Note* detailing the model and results, and included additional discussion in the *Manuscript*.

Figure R5. Thermal measurement and simulation results during the stability test under ~ 10 suns illumination. (a) Comparison between simulated and measured temperature rise in the bulk electrolyte at different pressures. (b) Simulated surface temperature of the photoelectrode after ~ 3.5 hours stability tests. Temperature distribution colormaps at the end of the test at (c) 1 bar and (d) 5 bar. During the stability measurement, undoped and uncatalyzed BiVO_4 was used as the photoanode, with a platinum mesh as the counter electrode (CE), and Ag/AgCl as the reference electrode. The PEC cells were operated potentiostatically at 1.23 V vs. RHE in a 0.1 M potassium phosphate (KPi) buffer solution ($\text{pH} = 7$), and an electrolyte flow rate of 4.6 mL s^{-1} was maintained during the stability measurements. A K-type thermocouple was used to measure the electrolyte temperature, according to the digital photographs of the measurement setup in **Fig. S10a**, *Supplementary Information*.

Associated changes to the Manuscript and Supplementary Information:

- *Results (Manuscript)*, page 21 – 22, line 372 – 393: “A heat transfer model was developed to evaluate the thermal effect associated with the prolonged operation under higher solar concentration, as detailed in *Supplementary Note S4* (see Fig. S16). The simulated results were validated against experimentally measured bulk electrolyte temperature, as shown in Fig. S17a. Despite its simplicity, this heat transfer model effectively captures the thermal trends observed during stability testing under ~10 suns illumination. Specifically, the measured bulk electrolyte temperature increased by approximately 5 °C at 1 bar and 3 °C at 5 bar under continuous illumination, while the model predicts a comparable rise of ~5 °C for both pressures. In addition, the simulation reveals that the maximum surface temperature rise of the photoelectrode can reach ~13 °C above ambient (Fig. S17b-S17d). While this value may be somewhat overestimated due to simplified boundary assumptions—namely, adiabatic treatment of the electrode sealing interfaces and the electrolyte boundary adjacent to the counter electrode—it nonetheless highlights the potential for localized heating at the photoelectrode surface. Such a temperature rise can indeed facilitate interfacial kinetics⁶⁴ and could potentially accelerate material degradation under extended operation.⁶⁵ These findings emphasize the importance of effective thermal management in high-pressure PEC systems under concentrated sunlight. Advanced cooling strategies, such as those demonstrations performed by Haussener and co-authors,^{9,11,25} can be implemented to prevent photoelectrode overheating under concentrated sunlight. Importantly, the consistent trends observed across under different pressures, both experimentally and in simulation, support the validity of our comparative performance analysis under elevated pressures.”
- *Supplementary Information*, page 26 – 32: the multiphysics model introduced above has been included as *Supplementary Note S4* in the revised version.
- *Supplementary Information*, page 29 – 32: **Figure R4** and **Figure R5** have now been added as **Fig. S16** and **Fig. S17** in the revised version, respectively.

5. It would be valuable for the authors to briefly discuss how the high-pressure PEC module might be integrated into a practical solar water splitting system.

Response: We sincerely thank the reviewer for this important suggestion. As noted in our response to Comment 4 of Reviewer #1, three possible strategies are proposed for storing or utilizing the high-pressure H₂ gas generated by the PEC water splitting cells, as illustrated schematically in **Fig. R1**. First, the produced H₂ can be stored in a high-pressure gas cylinders for transportation and later use. Second, it can be absorbed by the solid-state H₂ storage materials, such as MgNi, ZrCr, and LaNi₅-based metal hydrides. Third, the produced high-pressure H₂ can be directly fed into high-pressure hydrogen applications, such as hydrogenation reactors. This way, the elevated pressure can be maintained while collecting or directly utilizing the pressurized H₂. We have now added a brief discussion in the revised *Manuscript* and *Supplementary Information*.

Associated changes to the *Manuscript* and *Supplementary Information*:

- *Results (Manuscript)*, page 29, line 545 – 555: “In principle, the system can be self-pressurized via gas evolution during operation, potentially avoiding additional

energy input for pressurization other than the thermodynamic penalty of higher-pressure operation. In the pressure range of up to 20 bar, our recent analysis has shown that the benefits largely outweigh the thermodynamic penalty.³⁵ Nevertheless, we acknowledge that achieving and maintaining elevated pressure introduces technical and economic challenges, including the need for reinforced cell housings, thicker optical windows, enhanced sealing, pressure-rated tubing, and precise back-pressure control—all of which would increase capital and maintenance costs. A comprehensive techno-economic analysis is needed to evaluate the economic trade-offs and inform possible scale-up configurations.”

- *Supplementary Information, page 38: Figure R1* has now been added as **Fig. S22** in the revised version.

6. For better clarify the author’s point of view and attract more readers, it is recommended to add citations.(*Angew. Chem. Int. Ed.* 2020, 59, 23094-23099; *Nat. Sustain.*2025,doi.org/10.1038/s41893-025-01530-y; *Matter* 2024,7,2278-2293; *Angew. Chem. Int. Ed.* 2024,136,e202317414.).

Response: We appreciate the reviewer’s suggestion. The key conclusions of the recommended papers have now been integrated to enhance the *Discussion* section, and cited as Refs. 16, 17, 60, 62 in the revised *Manuscript*.

Associated changes to the *Manuscript*:

- *Results, page 28 – 29, line 522 – 533:* “First, in this initial demonstration of high-pressure PEC water splitting, only two representative and contrasting systems are selected: BiVO₄ as a metal oxide photoanode under back-illumination and a platinized triple-junction III-V photocathode under front-illumination. These photoelectrodes were intentionally chosen as benchmark photoelectrodes since they have been extensively studied in PEC water splitting research and beyond.^{16,17,66,73-77} They are relatively stable and have demonstrated adequately high photocurrent; this ensures that performance variations in our study can be attributed primarily to pressure-related effects rather than material-related factors, thus providing a reliable basis to assess their impact on device operation.”

Reviewer #3:

This manuscript demonstrates the operating photoelectrochemical (PEC) water splitting cells at elevated pressures (up to 8 bar), addressing a bottleneck in scalable solar hydrogen production. The authors developed a custom high-pressure flow cell and investigated two configurations: (1) a back-illuminated BiVO₄-based PEC cell, and (2) a front-illuminated integrated platinumized triple-junction (3J) III-V PEC device. For the BiVO₄ system, increased pressure suppressed gas bubble formation, resulting in up to a 50% improvement in photocurrent under 10 suns illumination. Operando imaging confirmed that the improved performance stemmed from reduced bubble coverage and size. In contrast, the 3J III-V PEC device exhibited pressure-independent photocurrent due to its long carrier diffusion length, thin dispersed Pt catalyst layer, and highly hydrophilic surface, which facilitated bubble detachment and minimized recombination losses. The manuscript underscores the configuration-dependent effects of pressure on PEC performance and highlights the benefit of high-pressure operation: directly generating pressurized hydrogen to reduce downstream compression energy costs. Overall, I think this is a great paper and the manuscript could meet the standard of Nature Communications after addressing the following issues.

Response: We sincerely appreciate the reviewer's positive feedback. We have carefully addressed all the listed comments/suggestions. The point-to-point responses are as follow:

1. The figure clarity should be improved. The authors need to make sure that the readers can gain the key breakthrough of this work by reading the introduction and figures. However, figures only contain the recorded data without highlighting the novelty of this work. For example, in Figure 1, they need to add Figure 1c to highlight the novelty as compared with the previous approach. Besides, there are too many labels in Figure 1b about the whole system, and I do not know how significance of this work as compared with other works. The same things happen in other figures, the authors need to improve the figure quality and highlight the key novelty.

Response: We sincerely thank the reviewer for this constructive suggestion. In the revised *Manuscript*, we have significantly improved figure clarity to better communicate the core innovation of our work, while assuring the formatting is consistent throughout the *Manuscript*. As an example, **Figure 1** has been redesigned in the revised version and is now presented as **Fig. R6**. A new schematic (**Fig. R6a**) has been added to clearly illustrate the research gap and motivation for operating PEC water splitting devices at elevated pressure. The original system overview (previously **Fig. 1b**) has been moved to the *Supplementary Information* (now **Fig. S1**) to reduce visual complexity and enhance readability in the main text.

Additionally, we have carefully revised the figure annotations across the *Manuscript* to better highlight the novelty and significance of our findings, as suggested. We hope these changes will improve the overall clarity and impact of the figures.

Figure R6. (a) Motivation for operating PEC water splitting devices at elevated pressure, and (b) schematic illustration of the high-pressure flow cell (HPFC) developed in this work. In (a), a typical PEC water splitting setup is shown on the left, highlighting that all demonstrations to date operate under atmospheric pressure. In contrast, most downstream hydrogen applications—such as fuel cells, hydrogenation processes, and ammonia synthesis—require high-pressure hydrogen, as indicated on the right. This reveals a critical research gap between current academic PEC studies and industrial-scale requirements. Panel (b) presents the schematic of the HPFC designed, constructed, and tested in this study. An exploded technical view is provided in Fig. S1, and the digital photographs of the assembled HPFC are shown in Fig. S2. The electrolyte is circulated by a rotary pump, achieving laminar flow between the electrodes via an optimized flow distributor (see Fig. S2c). The system is pressurized using external gas (N₂ or O₂, depending on the reaction), and the operating pressure is regulated by a back-pressure controller. Simulated AM1.5G illumination—with optional Fresnel lenses—enables solar concentrations up to ~10 suns. Electrolyte temperature is monitored at the outlet via a K-type thermocouple. The downstream pressure requirements for hydrogen infrastructure are adapted from Referecne²⁰. For clarity, standard fittings (e.g., nuts, connectors, PTFE tubing) are omitted in the schematic.

Associated changes to the *Manuscript* and *Supplementary Information*:

- *Results (Manuscript)*, page 7 – 8: **Figure R6** has now been added as **Fig. 1** in the revised version.
 - *Supplementary Information*, page 2 – 3: **Figure 1b** in the previous version (*Manuscript*) has now been added as **Fig. S1** in the revised version.
2. To add pressure for hydrogen production, the long-term stability and safety issues require a second consideration. During large-scale applications, will the materials degrade under cyclic pressure/continuous pressure? Is there any auxiliary equipment required for pressure adding, and will this increase the total cost of the hydrogen production?

Response: We thank the reviewer for this valuable comment. We have addressed the concern regarding long-term stability in our responses to Comment 6 of Reviewer #1 and Comment 3 of Reviewer #2 by conducting additional long-term stability measurements on the back-illuminated BiVO₄-based PEC cell under varying pressures. Chronoamperometry (CA) tests were conducted under potentiostatic conditions at 1.23 V vs. RHE, in a 0.1 M potassium phosphate buffer (KPi, pH = 7) with an electrolyte flow of 4.6 mL s⁻¹. Two illumination intensities were applied: 1 sun (AM1.5G) and 10 suns. The CA test at 1 sun was terminated when the photocurrent dropped by ~50%; while under 10 suns, the test was stopped at ~100% photocurrent loss. Since uncatalyzed and undoped BiVO₄ samples were used in our study, photocurrent decrease is expected;^{5,6} however, we emphasize that the focus here is to understand if pressure elevation introduces any effect on degradation. Interestingly, as shown in **Fig. R2**, no significant difference is observed in the normalized photocurrent plots at different pressure, both at 1 and 10 suns. This suggests that the BiVO₄ degradation mechanism is the same at different pressures. Overall, our study does not suggest any presence of pressure-induced degradation, but further detailed investigations are required to reveal the mechanisms. Finally, within the pressure range investigated, no gas or liquid leakage was detected during prolonged high-pressure operation, confirming the mechanical integrity and operational robustness of the system. This discussion has now been included in the *Manuscript* and the *Supplementary Information*.

Regarding the reviewer's comment on the need of further auxiliary equipment, we have also addressed this in our response to Comment 3 of Reviewer #1. Reducing the need for downstream hydrogen compression and suppression of bubble-induced losses represent significant benefits of operating PEC water splitting cells at elevated pressure. In our experiments, we supplied compressed gas to the cell for controlled conditions; however, in practical applications, the system can be self-pressurized through gas production during operation. At such, no additional energy input is required for system pressurization other than the thermodynamic penalty of higher-pressure operation. Our recent analysis shows that the benefits still largely outweigh this thermodynamic penalty in the pressure range of up to 20 bar.¹ Nevertheless, we fully acknowledge that increasing the operating pressure may also introduce nontrivial technical and economic challenges. These include the need for thicker glass components, more robust sealing strategies to prevent electrolyte and gas

leakage, and the use of high-pressure-rated tubing and control systems—all of which can increase the capital and maintenance costs. Detailed techno-economic assessment beyond the scope of our current study is thus necessary to comprehensively evaluate the economic feasibility of operating PEC systems at elevated pressures. We have now added a brief discussion regarding the points above in the revised *Manuscript*.

Associated changes to the *Manuscript* and *Supplementary Information*:

- *Results (Manuscript)*, page 21, line 362 – 371: “To investigate whether pressure elevation introduces any effect on the stability of the back-illuminated BiVO₄ photoanode, potentiostatic experiments were performed with the same PEC cell at 1.23 V vs. RHE. Illumination intensities of 1 sun and 10 suns were applied. As shown in Fig. S15, the photocurrent decreases with time, indicating degradation; this is expected since our BiVO₄ is uncatalyzed and has no protection layer.^{25,63} However, the photocurrent decrease remains similar at different pressures, suggesting that pressure elevation has minimal impact on the degradation behavior of the BiVO₄ photoelectrode. Moreover, no gas or liquid leakage was observed during the extended high-pressure operation, indicating reliable structural mechanical integrity and operational stability of the system.”
 - *Results (Manuscript)*, page 29, line 537 – 555: “On the system level, the benefits from operating PEC water splitting cells at elevated pressure should be ensured to outweigh the associated penalties. In addition to the suppression of bubble-induced losses, reducing or eliminating the need for downstream hydrogen compression represents a significant advantage. In practice, the elevated pressure PEC cell can be connected directly to high-pressure H₂ tanks, solid-state H₂ storage materials, or any applications that require elevated pressure operation (e.g., Haber-Bosch or hydrogenation reactors), as illustrated in Fig. S22. For instance, when compressing H₂ to 875 bar, doing so by starting from 1 bar consumes ~80% more energy than starting from 20 bar (~4.8 vs. ~2.7 kWh kg⁻¹).⁷⁸⁻⁸⁰ In principle, the system can be self-pressurized via gas evolution during operation, potentially avoiding additional energy input for pressurization other than the thermodynamic penalty of higher-pressure operation. In the pressure range of up to 20 bar, our recent analysis has shown that the benefits largely outweigh the thermodynamic penalty.³⁵ Nevertheless, we acknowledge that achieving and maintaining elevated pressure introduces technical and economic challenges, including the need for reinforced cell housings, thicker optical windows, enhanced sealing, pressure-rated tubing, and precise back-pressure control—all of which would increase capital and maintenance costs. A comprehensive techno-economic analysis is needed to evaluate the economic trade-offs and inform possible scale-up configurations.”
 - *Supplementary Information*, page 25: **Figure R2** has now been added as **Fig. S15** in the revised version.
3. The authors focused more on the breakthrough at the systematic level, and the materials used are simple. However, I still want the authors to clarify why BiVO₄ is applied in this system. What is the key advantage as compared with other catalysts?

Besides, instead of only simple materials characterization in supplementary information, is there any difference/breakthrough on the materials level?

Response: We thank the reviewer for the thoughtful comment. Our primary focus in the present study is to investigate the impact of operating pressure and gas bubble dynamics on the overall performance of PEC water splitting devices, rather than introducing breakthroughs at the materials level. Accordingly, BiVO₄ was intentionally selected as a model photoanode due to its well-established performance, relative stability, and extensive characterization in the literature. BiVO₄ is widely regarded as one of the highest-performing and most studied complex oxide photoelectrodes for PEC water oxidation. The use of benchmark material like BiVO₄ allows for a clearer attribution of performance changes to pressure-related effects, rather than to changes in material properties. We have clarified this point in the revised *Manuscript* to prevent potential confusion.

Associated changes to the *Manuscript*:

- **Results (*Manuscript*), page 28 – 29, line 521 – 536:** “We will now discuss some general limitations of our study and provide an outlook for operating PEC devices at elevated pressure. First, in this initial demonstration of high-pressure PEC water splitting, only two representative and contrasting systems are selected: BiVO₄ as a metal oxide photoanode under back-illumination and a platinized triple-junction III-V photocathode under front-illumination. These photoelectrodes were intentionally chosen as benchmark photoelectrodes since they have been extensively studied in PEC water splitting research and beyond.^{16,17,66,73-77} They are relatively stable and have demonstrated adequately high photocurrent; this ensures that performance variations in our study can be attributed primarily to pressure-related effects rather than material-related factors, thus providing a reliable basis to assess their impact on device operation. At the same time, expanding this comparison to additional semiconductors, such as α-Fe₂O₃ and WO₃ photoanodes or Sb₂S₃ and Cu₂O photocathodes, would offer further insights into any material-specific pressure effects. These studies are part of our ongoing efforts and will be reported in due course.”

4. Please expand the discussion on pressure-dependent bubble nucleation dynamics. The manuscript attributes performance gains to suppressed bubble evolution at higher pressure, but the underlying physical mechanism (e.g., Laplace pressure, critical bubble size, contact angle effects) should be further elaborated.

Response: We appreciate the reviewer’s insightful comment on pressure-dependent bubble nucleation dynamics. As noted in our response to Comment 2 of Reviewer #2, we have conducted additional imaging experiments using a macroscopic shadowgraphy setup to observe oxygen bubble evolution on the BiVO₄ photoanode surface (**Fig. R3**). Representative images (**Fig. R3b**) clearly show that increasing pressure significantly suppresses gas bubble formation on the BiVO₄ surface, consistent with localized observations presented in **Fig. 4e–f** of the *Manuscript*. At 1 bar, bubbles were randomly distributed across the porous BiVO₄ layer, with larger bubble sizes observed near the top of the electrode—which can be attributed to

hydrostatic pressure gradients. However, bubble presence was already negligible at pressures above 2 bar. This insight should be considered for the scale-up design of PEC systems.

We also appreciate the reviewer's suggestion to investigate pressure-dependent bubble nucleation dynamics. Unfortunately, direct observation of nucleation events was not feasible due to the limited resolution of our current imaging setup. However, related studies, such as a recent work on bubble evolution on TiO₂ micro-photoelectrodes,⁷ confirm that elevated pressure facilitates bubble detachment due to increased dissolved gas saturation—aligned with previous simulation results.⁸ Finally, we emphasize that the topic of gas bubble dynamics attracts increasing interests from researchers in the field, and a full understanding of this complex dynamics under pressure remains an open question. Advanced experimental and theoretical techniques such as atomic force microscopy⁹⁻¹¹ and molecular dynamics simulations^{12,13} will be essential for future investigation.

Additional discussion has now been added to the revised *Manuscript* and *Supplementary Information*.

Associated changes to the *Manuscript* and *Supplementary Information*:

- *Results (Manuscript)*, page 20, line 352 – 361: “To further elucidate the effects of pressure on gas bubble evolution throughout the whole surface of the 2 cm × 2 cm BiVO₄ photoelectrode, we conducted macroscopic imaging of oxygen bubbles. The experimental setup was modified accordingly (Fig. S14a) and introduced in the *Methods* section. At 1 bar, bubbles appeared randomly across the porous BiVO₄ surface (Fig. S14b), with larger bubble sizes observed near the top of the electrode; we tentatively attribute this observation to the presence of hydrostatic pressure gradients. This insight is useful for the scale-up of PEC devices, as non-uniform bubble detachment may result in spatial variations in performance. However, at pressures above 2 bar, bubble presence was already negligible (Fig. S14b). This observation is consistent with our localized observations shown in Fig. 4 e-f.”
- *Results (Manuscript)*, page 17, line 315 – 333: “Although bubble nucleation events could not be directly resolved, recent work on bubble evolution on TiO₂ micro-photoelectrodes showed that elevated pressure facilitates bubble detachment, primarily due to higher dissolved gas saturation.⁴⁸ Consistent with these findings, our measurements (Fig. 4e–f) revealed smaller and fewer bubbles under elevated pressure. This behavior can be tentatively explained by the formation of a supersaturated boundary layer (SSBL) of dissolved gas near the (photo)electrode at higher pressure. Our recent simulation showed that the SSBL facilitates bubble detachment from the surface of the (photo)electrode via Marangoni convection.⁴⁹ Additional contributing factors may include an increased Laplace pressure and a larger critical radius for nucleation at higher pressures, both of which inhibit initial bubble formation. Interfacial properties, such as the gas–liquid contact angle, may also play a role in pressure-dependent detachment dynamics, but direct experimental measurement of contact angles and nano/micro bubbles on semiconductor surfaces (e.g., BiVO₄ photoelectrode) under elevated pressure

remains technically challenging. High-resolution tools such as atomic force microscopy and spectroscopy techniques⁵⁰⁻⁶⁰ and molecular dynamics simulation^{61,62} will be essential, which are beyond the scope of this study. Furthermore, the porous morphology of our BiVO₄ samples (see Fig. S6) introduces additional complexity, as it may lead to random spatial distribution of nanobubbles, making their detection even more complicated.”

- *Methods (Manuscript)*, page 38 – 39, line 757 – 766: “To enable full-field visualization of the bubble evolution on the BiVO₄ photoanode (2 cm × 2 cm), the setup was further modified as shown in the schematic illustration in Fig. S14a. The same BiVO₄ and Ag/AgCl electrodes were kept as the working and reference electrodes, but the counter electrode was a platinum wire instead of the platinum mesh to reduce visual obstruction. The PEC cell was operated at 0.5 mA cm⁻² in static 0.1 M KP_i (pH = 7), and bubble distribution was recorded using a Zeiss Makro-Planar T* 2/100 lens coupled to a high-speed camera (HS5-Q, 2560 × 2048 pixels), with a resolution limit of ~45–50 μm in bubble diameter. The representative images of bubbles on the BiVO₄ surface under different pressures are shown in Fig. S14b.”
- *Supplementary Information*, page 23 – 24: **Figure R3** has now been added as **Fig. S14** in the revised version.

5. The observation of bubbles plays a significant role in understanding the reason for high water splitting. Therefore, I recommend improving the bubble imaging resolution. Can you specify the optical resolution limits of your cameras, and whether any contrast-enhancing techniques (e.g., DIC, phase contrast) were attempted? Suggest future tools (e.g., AFM, total internal reflection microscopy) to visualize nanobubbles.

Response: We appreciate the reviewer’s valuable comment and constructive suggestion. In our current study, two imaging setups were employed: a high-speed microscopic imaging system using a telescopic lens for localized observation (as in **Fig. 4e–f, Manuscript**), and a macroscopic imaging system using a Zeiss Makro-Planar T* 2/100 lens to capture bubble evolution over the entire 2 cm × 2 cm BiVO₄ electrode surface (as in **Fig. S14b**). The macroscopic system, coupled with a 2560 × 2048 pixel high-speed camera (HS5-Q), provides an estimated spatial resolution of ~45–50 μm. The localized telescopic setup achieves higher resolution (~10 μm), albeit over a smaller field of view (~5 mm × 4 mm). These values define the current optical detection thresholds in our experiments.

Due to hardware constraints and the use of thick optical windows in the pressurized cell, contrast-enhancing techniques such as Differential Interference Contrast (DIC) or phase contrast microscopy were not feasible. These methods require specialized objectives, precise optical alignment, and short working distances, all of which are incompatible with our sealed high-pressure PEC system.

To enhance image clarity, we applied post-processing techniques in ImageJ, including background subtraction and contrast normalization. These helped improve visual distinction of gas bubbles, although lighting non-uniformities and optical limitations still affected the overall image quality. Nevertheless, since the experimental

conditions are consistent across all tests, relative comparisons between our experimental results remain valid.

We fully agree that future investigations into nanobubble dynamics will benefit from higher-resolution tools. Techniques such as atomic force microscopy (AFM), total internal reflection microscopy (TIRM), and high-speed confocal imaging are under active consideration in our ongoing work to explore nucleation, growth, and detachment processes under pressure.

We have now clarified these limitations and future directions in the revised *Manuscript*.

Associated changes to the *Manuscript* and *Supplementary Information*:

- *Results (Manuscript)*, page 20, line 352 – 361: “To further elucidate the effects of pressure on gas bubble evolution throughout the whole surface of the 2 cm × 2 cm BiVO₄ photoelectrode, we conducted macroscopic imaging of oxygen bubbles. The experimental setup was modified accordingly (Fig. S14a) and introduced in the *Methods* section. At 1 bar, bubbles appeared randomly across the porous BiVO₄ surface (Fig. S14b), with larger bubble sizes observed near the top of the electrode; we tentatively attribute this observation to the presence of hydrostatic pressure gradients. This insight is useful for the scale-up of PEC devices, as non-uniform bubble detachment may result in spatial variations in performance. However, at pressures above 2 bar, bubble presence was already negligible (Fig. S14b). This observation is consistent with our localized observations shown in Fig. 4 e-f.”
- *Methods (Manuscript)*, page 38 – 39, line 737 – 766: “Two complementary imaging setups were used to observe gas bubble evolution on the electrodes from different angles during operation, as shown by the schematic in Fig. S6. For the back-illuminated PEC cell, a high-speed camera (HS5-Q Quad HD+ Resolution (2560 × 2048 pixels) with a Navitar Zoom 6000 Lens System, FasTec) was used to capture the side-view of gas bubble evolution, and the shadowgraphy of bubbles on the electrodes was captured using a LaVision camera system (Imager SX 6M CCD camera, 2752 × 2200 pixels). For the front-illuminated PEC cell, only the high-speed camera was used to observe the side-view of gas bubbles. The camera was fixed higher than the 3J III-V photoelectrode so that the surface of the sample was visible. Neutral density (ND) filters with various ND factors were used depending on the light condition to enhance the imaging quality. The localized telescopic imaging configuration achieved a spatial resolution of approximately 10 μm over a limited field of view (~5 mm × 4 mm), suitable for capturing detailed bubble dynamics in selected regions of interest (see Fig. 4e-f). The images were processed using ImageJ, with additional validation using DaVis 10 (LaVision). To enhance image clarity, we applied post-processing techniques in ImageJ, including background subtraction and contrast normalization. These helped improve visual distinction of gas bubbles, although lighting non-uniformities and optical limitations still affected the overall image quality. Nevertheless, since the experimental conditions are consistent across all tests, relative comparisons between our experimental results remain valid.”

To enable full-field visualization of the bubble evolution on the BiVO₄ photoanode (2 cm × 2 cm), the setup was further modified as shown in the schematic illustration in Fig. S14a. The same BiVO₄ and Ag/AgCl electrodes were kept as the working and reference electrodes, but the counter electrode was a platinum wire instead of the platinum mesh to reduce visual obstruction. The PEC cell was operated at 0.5 mA cm⁻² in static 0.1 M KP_i (pH = 7), and bubble distribution was recorded using a Zeiss Makro-Planar T* 2/100 lens coupled to a high-speed camera (HS5-Q, 2560 × 2048 pixels), with a resolution limit of ~45–50 μm in bubble diameter. The representative images of bubbles on the BiVO₄ surface under different pressures are shown in Fig. S14b.”

6. Since this technique requires 10 sun for water splitting, the authors need to consider possible heat accumulation at high solar flux. Could longer operation at 10 suns lead to cumulative heating or catalyst degradation?

Response: We thank the reviewer for rising this important issue. As noted in our response to Comment 4 of Reviewer #2, we conducted additional simulations and experiments to monitor the thermal effect of continuous illumination of ~10 suns on the BiVO₄ photoelectrode. The heat transfer model is described in **Fig. R4** and the results are shown in **Fig. R5**.

In brief, the simulated results were validated against experimentally measured bulk electrolyte temperature, as shown in **Fig. R5a**. Despite its simplicity, this heat transfer model effectively captures the thermal trends observed during stability testing under ~10 suns illumination. Specifically, the measured bulk electrolyte temperature increased by approximately 5 °C at 1 bar and 3 °C at 5 bar under continuous illumination, while the model predicts a comparable rise of ~5 °C for both pressures. In addition, the simulation reveals that the maximum surface temperature rise of the photoelectrode can reach ~13 °C above ambient (**Fig. R5b-R5d**). While this estimate may be somewhat overestimated due to simplified boundary assumptions—namely, adiabatic treatment of the electrode sealing interfaces and the electrolyte boundary adjacent to the counter electrode—it nonetheless highlights the potential for localized heating at the photoelectrode surface. Such a temperature rise can enhance interfacial kinetics¹⁸ and could potentially accelerate material degradation under extended operation¹⁴. These findings emphasize the importance of effective thermal management in high-pressure PEC systems under concentrated sunlight. Advanced cooling strategies, such as those demonstrated in Ref.¹⁹ and Ref.¹⁷, can be implemented to prevent photoelectrode overheating under concentrated sunlight. Importantly, the consistent trends observed across under different pressures, both experimentally and in simulation, support the validity of our comparative performance analysis under elevated pressures.

We have now addressed and discussed this point in the revised *Manuscript* and *Supplementary Information*.

Associated changes to the *Manuscript* and *Supplementary Information*:

- *Results (Manuscript)*, page 21 – 22, line 372 – 393: “A heat transfer model was developed to evaluate the thermal effect associated with the prolonged operation under higher solar concentration, as detailed in *Supplementary Note S4* (see Fig. S16). The simulated results were validated against experimentally measured bulk electrolyte temperature, as shown in Fig. S17a. Despite its simplicity, this heat transfer model effectively captures the thermal trends observed during stability testing under ~10 suns illumination. Specifically, the measured bulk electrolyte temperature increased by approximately 5 °C at 1 bar and 3 °C at 5 bar under continuous illumination, while the model predicts a comparable rise of ~5 °C for both pressures. In addition, the simulation reveals that the maximum surface temperature rise of the photoelectrode can reach ~13 °C above ambient (Fig. S17b-S17d). While this value may be somewhat overestimated due to simplified boundary assumptions—namely, adiabatic treatment of the electrode sealing interfaces and the electrolyte boundary adjacent to the counter electrode—it nonetheless highlights the potential for localized heating at the photoelectrode surface. Such a temperature rise can indeed facilitate interfacial kinetics⁶⁴ and could potentially accelerate material degradation under extended operation.⁶⁵ These findings emphasize the importance of effective thermal management in high-pressure PEC systems under concentrated sunlight. Advanced cooling strategies, such as those demonstrations performed by Haussener and co-authors,^{9,11,25} can be implemented to prevent photoelectrode overheating under concentrated sunlight. Importantly, the consistent trends observed across under different pressures, both experimentally and in simulation, support the validity of our comparative performance analysis under elevated pressures.”
- *Supplementary Information*, page 26 – 32: the multiphysics model introduced above has been included as *Supplementary Note S4* in the revised version.
- *Supplementary Information*, page 29 – 32: **Figure R4** and **Figure R5** have now been added as **Fig. S16** and **Fig. S17** in the revised version, respectively.

Reviewer #4:

Response: We sincerely appreciate the reviewer's effort and time for reviewing our work. We have carefully addressed all the listed comments/suggestions. We hope that this revised version of our *Manuscript and Supplementary Information* will be considered favourably.

References

- 1 Liang, F., van de Krol, R. & Abdi, F. F. Assessing elevated pressure impact on photoelectrochemical water splitting via multiphysics modeling. *Nature Communications* **15**, 4944 (2024).
- 2 Obata, K., Mokeddem, A. & Abdi, F. F. Multiphase fluid dynamics simulations of product crossover in solar-driven, membrane-less water splitting. *Cell Reports Physical Science* **2**, 100358 (2021).
- 3 Swiegers, G. F. *et al.* Current status of membraneless water electrolysis cells. *Current Opinion in Electrochemistry* **32**, 100881, doi:<https://doi.org/10.1016/j.coelec.2021.100881> (2022).
- 4 Schalenbach, M. *et al.* Gas permeation through nafion. Part 1: measurements. *The Journal of Physical Chemistry C* **119**, 25145-25155 (2015).
- 5 Toma, F. M. *et al.* Mechanistic insights into chemical and photochemical transformations of bismuth vanadate photoanodes. *Nature Communications* **7**, 12012, doi:10.1038/ncomms12012 (2016).
- 6 Holmes-Gentle, I., Bedoya-Lora, F. E., Aimone, L. & Haussener, S. Photoelectrochemical behaviour of photoanodes under high photon fluxes. *Journal of Materials Chemistry A* **11**, 23895-23908 (2023).
- 7 Luo, X., Xu, Q., Nie, T., She, Y. & Guo, L. Effect of pressurization on bubble dynamics of photoelectrochemical water splitting. *Physics of Fluids* **36** (2024).
- 8 Liang, F., van de Krol, R. & Abdi, F. F. The influence of dissolved gas supersaturation on bubble detachment from planar (photo) electrodes. *Cell Reports Physical Science* **5** (2024).
- 9 Nellist, M. R. *et al.* Potential-sensing electrochemical atomic force microscopy for in operando analysis of water-splitting catalysts and interfaces. *Nature Energy* **3**, 46-52 (2018).
- 10 Esposito, D. V. *et al.* Methods of photoelectrode characterization with high spatial and temporal resolution. *Energy & Environmental Science* **8**, 2863-2885 (2015).
- 11 Yu, J. *et al.* Interfacial nanobubbles' growth at the initial stage of electrocatalytic hydrogen evolution. *Energy & Environmental Science* **16**, 2068-2079 (2023).
- 12 Zhan, S. *et al.* Investigation of electrolytic hydrogen nanobubbles behavior on heterogeneous wettability surface by using molecular dynamics simulation. *International Journal of Hydrogen Energy* **112**, 160-171 (2025).
- 13 Xi, W. *et al.* Electrocatalytic generation and tuning of ultra-stable and ultra-dense nanometre bubbles: an in situ molecular dynamics study. *Nanoscale* **13**, 11242-11249 (2021).
- 14 Bedoya-Lora, F. E., Holmes-Gentle, I., Feurstein, P. & Haussener, S. Effect of Operating Conditions on the Degradation of BiVO₄ Photoanodes. *Advanced Functional Materials*, 2505102.
- 15 Cao, S. *et al.* Ultrasmall CoP nanoparticles as efficient cocatalysts for photocatalytic formic acid dehydrogenation. *Joule* **2**, 549-557 (2018).
- 16 Xiao, M., Wang, Z., Maeda, K., Liu, G. & Wang, L. Addressing the stability challenge of photo (electro) catalysts towards solar water splitting. *Chemical Science* **14**, 3415-3427 (2023).

- 17 Holmes-Gentle, I., Tembhrne, S., Suter, C. & Haussener, S. Kilowatt-scale solar hydrogen production system using a concentrated integrated photoelectrochemical device. *Nature Energy*, 1-11 (2023).
- 18 Zhang, L. *et al.* Significantly enhanced photocurrent for water oxidation in monolithic Mo: BiVO₄/SnO₂/Si by thermally increasing the minority carrier diffusion length. *Energy & Environmental Science* **9**, 2044-2052 (2016).
- 19 Tembhrne, S., Nandjou, F. & Haussener, S. A thermally synergistic photoelectrochemical hydrogen generator operating under concentrated solar irradiation. *Nature Energy* **4**, 399-407 (2019).
- 20 Ivanova, M. E. *et al.* Technological Pathways to Produce Compressed and Highly Pure Hydrogen from Solar Power. *Angewandte Chemie International Edition*, e202218850 (2023).

Response to reviewers' comments

Manuscript ID: NCOMMS-25-32194A

Title: Operating photoelectrochemical water splitting cells at elevated pressure: a demonstration using BiVO₄ and platinized III-V semiconductor photoelectrodes

In this response letter, the reviewers' comments to our original manuscript are provided in **black**, and our point-by-point responses as well as the corresponding changes to the manuscript are shown in **blue**.

Reviewers' Comments:

Reviewer #1:

The authors have addressed all comments from reviewers and the manuscript can be accepted for publication in the current version.

Response: We sincerely appreciate the reviewer's effort in reviewing our work and providing the positive feedback. We are glad to know that the current version of our manuscript is acceptable for publication.

Reviewer #2:

The author has already provided excellent solutions to the previously raised questions, but there are a few minor points below that require clarification.

Response: We sincerely appreciate the reviewer's positive feedback. We have carefully addressed all the additional minor comments/suggestions point-by-point below:

1. Could the author elaborate on how the maximum pressure value measured in the experimental setup was determined, and whether this value was constrained by safety considerations?

Response: We appreciate the reviewer's practical question. During our experiments, the operating pressure was measured using a back-pressure controller (Bronkhorst High-Tech, uncertainty: 0.2%). The maximum pressure achieved was 8 bar(a), and higher pressure has not been tested to comply with our current safety guidelines.

To avoid further confusion, this information has now been further clarified in our revised *Manuscript*.

Associated changes to the Manuscript:

- *Results (Manuscript)*, page 5, line 111 – 119: "High pressure operation was accomplished by supplying compressed gas into the cell while regulating the outflow rate of gases with a back-pressure controller (Bronkhorst High-Tech, uncertainty: 0.2%). Note that this approach to regulate the pressure was chosen because it offers easy control and experimental convenience in a lab-based

setting; in practical applications, PEC water splitting cells can self-pressurize through gas production during operation, requiring only a pressure relieve valve as a control mechanism. Prior to PEC experiments, the setup was tested for safe operation up to 8 bar(a); higher pressures have not been tested to comply with current safety guidelines.”

2. What methodology was employed to determine the effective electrode area used in the photoelectrochemical (PEC) tests?

Response: For BiVO₄ photoelectrodes, the effective area was determined by the size of the illumination windows, which is 4 cm² (visible in the inset in Fig. R1). For the platinized 3J III-V photoelectrodes, the effective area was measured by analyzing digital photographs of the photoelectrodes in ImageJ. The digital photographs included a portion of a ruler so that the scale could be set, as shown in Fig. R2. This information has now been clarified in the revised *Manuscript* and *Supplementary Information*.

Figure R1. Exploded view of the high-pressure flow cell (HPFC) developed for this study. The electrolyte (not shown) is circulated using a rotary pump, establishing laminar flow between the electrodes via an optimized flow distributor (see Fig. S2c). The system is pressurized with external gas (N₂ or O₂, depending on the reaction), and the operating pressure is regulated by a back-pressure controller. Simulated AM1.5G illumination—optionally concentrated using Fresnel lenses—enables solar intensities up to ~10 suns. The area of the illumination window (visible in the inset) is ~4 cm², which determines the maximum effective area of the photoelectrodes. Observation windows are fabricated from 5 mm-thick toughened glass, allowing direct visualization of gas bubble evolution during operation at elevated pressure up to 8 bar. Standard fittings (e.g., nuts, connectors, PTFE tubing) are omitted in the schematic for clarity.

[FIGURE REDACTED]

Figure R2. Measurement of the effective area of the platinized 3J III-V photoelectrode. The opensource software ImageJ was used, while the scale was calibrated with the ruler visible in the digital photograph. For this electrode, the measured effective area was 0.66 cm².

Associated changes to the Manuscript and Supplementary Information:

- *Methods (Manuscript)*, page 29, line 598 – 602: “The reactor part was 3D printed using stainless steel. A pair of sample holders can be installed, which allows a “face-to-face” arrangement of two 2 cm × 2 cm (maximum) samples with a distance of ~4 mm. The maximum effective area of the photoelectrodes is therefore ~4 cm², which is constrained by the size of the illumination window (see Fig. S1).”
- *Methods (Manuscript)*, page 35, line 666 – 679: “Samples were prepared for the PEC experiment in a “lollipop” format. Ag conductive tape was applied lengthwise on the middle of glass substrate. To ensure a good ohmic contact, conductive Ag epoxy (Circuitworks CW2400, Chemtronics) was applied between Ag conductive tape and the back-side metallization of the prepared III-V cell. The conductive epoxy was cured on a hot plate at 110 °C for 10 min. The edges were covered using a black insulating two-part silicone resin (101RF, Microset Products Ltd, see Fig. S18) and the remaining electrochemically active surface was determined from digital images using the open-source software ImageJ, as shown in Fig. S18d. The resulting photoactive area of the sample is ~0.6 – 0.7 cm². The fabricated photoelectrode is shown in Fig. S18.”
- *Supplementary Information: Figure S1* has now been updated with **Fig. R1** in the revised version.
- *Supplementary Information: Figure R2* has now been added as **Fig. S18d** in the revised version.

3. How does variation in pressure influence the onset potential of the photoelectrode?

Response: We thank the reviewer for this insightful question. The effect of pressure on the onset potential can already be observed in Fig. S11 (reproduced as Fig. R3 below) and Figs. 5b-c (reproduced as Fig. R4 below). Overall, pressure change did not lead to a noticeable impact to the onset potential of either the BiVO₄ or the platinized 3J III-V photoelectrodes. To further support this observation, we conducted additional linear sweep voltammetry (LSV) measurements with BiVO₄ photoelectrode under chopped-light condition (1 sun). As shown in the LSV curves in Fig. R5, the onset potential of the photoelectrode remains unchanged with pressure variation. This information has now been clarified in the revised *Manuscript* and *Supplementary Information*.

Figure R3. (a) LSV scans of the back-illuminated BiVO₄ photoelectrode at different pressure under AM1.5G ($c = 1$), the electrolyte was a 0.1 M KP_i (pH = 7) solution. (b) and (c) show the same results as in panel (a) but under concentration of $c \approx 3$ and $c \approx 10$, respectively. Additional measurement at 1 bar (dashed curve) was performed after that at 8 bar to check for reproducibility. (d) Same as (a) but with the addition of the hole scavenger (0.5 M Na₂SO₃). The compressed O₂ gas was used for the electrolyte purging and cell pressurization. The scan rate for the LSV was 20 mV s⁻¹ and the electrolyte was circulated by a rotary pump at a constant flow rate of approximately 4.6 mL s⁻¹.

Figure R4. Linear sweep voltammetry (LSV) curves of the platinized 3J III-V photoelectrode measured under AM1.5G 1 sun illumination at different pressures (a) in a 0.1 M KPi solution ($\text{pH} = 7$) and (b) in a 0.1 M KPi solution containing 0.5 M $\text{Na}_2\text{S}_2\text{O}_8$ as electron scavenger.

Figure R5. Chopped-light (1 sun) J - V characteristics of BiVO_4 photoelectrode at different pressures, measured at a scan rate of 20 mV s^{-1} in 0.1 M KPi ($\text{pH} = 7$). The onset potentials for each curve are indicated with the red circles. Compressed O_2 gas was used for electrolyte purging and cell pressurization. The electrolyte was circulated by a rotary pump at a constant flow rate of approximately 4.6 mL s^{-1} .

Associated changes to the Manuscript and Supplementary Information:

- *Results (Manuscript), page 15, line 275 – 282: “At 1 sun, the photocurrent remains nearly constant at $\sim 0.5 \text{ mA cm}^{-2}$ across the pressure range of 1 – 8 bar. At ~ 3 suns, however, the photocurrent increases from $\sim 0.9 \text{ mA cm}^{-2}$ at 1 bar to $\sim 1.2 \text{ mA cm}^{-2}$ at 5 bar. The effect is more pronounced at ~ 10 suns, where the photocurrent increases from $\sim 1.7 \text{ mA cm}^{-2}$ at 1 bar to $\sim 2.5 \text{ mA cm}^{-2}$ at 5 bar. Furthermore, as shown in Fig. S11, the onset potential of the BiVO_4 photoelectrode remains unchanged with varying operating pressure. These findings demonstrate a key advantage of operating PEC water splitting devices under elevated pressure, particularly at higher solar concentrations.”*
- *Supplementary Information: **Figure R5** has now been added as **Fig. S11e** in the revised version.*

Reviewer #3:

I am happy with the revision and the manuscript could be accepted in its current form.

Response: We sincerely appreciate the reviewer’s effort in reviewing our work and providing the positive feedback. We are glad to hear that the current version of our manuscript is acceptable for publication.

Reviewer #4:

Response: We sincerely appreciate the co-reviewer’s positive feedback and effort in evaluating our work.